# MOSAIC: Multi-Subject Personalized Generation via Correspondence-Aware Alignment and Disentanglement

**Dong She**[1][*], **Siming Fu**[1][*][†], **Mushui Liu**[2][*], **Qiaoqiao Jin**[1][*], **Hualiang Wang**[1,3],
**Mu Liu**[1], **Jidong Jiang**[1][‡]
[1]ByteDance   [2]Zhejiang University   [3]The Hong Kong University of Science and Technology
`{shedong, fusiming, jinqiaoqiao, liumu, jiangjidong}@bytedance.com,`
`lms@zju.edu.cn, hwangfd@connect.ust.hk`

## Abstract

Multi-subject personalized generation presents unique challenges in maintaining identity fidelity and semantic coherence when synthesizing images conditioned on multiple reference subjects. Existing methods often suffer from identity blending and attribute leakage due to inadequate modeling of how different subjects should interact within shared representation spaces. We present **MOSAIC**, a representation-centric framework that rethinks multi-subject generation through explicit semantic correspondence and orthogonal feature disentanglement. Our key insight is that multi-subject generation requires precise semantic alignment at the representation level—knowing exactly which regions in the generated image should attend to which parts of each reference. To enable this, we introduce SemAlign-MS, a meticulously annotated dataset providing fine-grained semantic correspondences between multiple reference subjects and target images, previously unavailable in this domain. Building on this foundation, we propose the semantic correspondence attention loss to enforce precise point-to-point semantic alignment, ensuring high consistency from each reference to its designated regions. Furthermore, we develop the multi-reference disentanglement loss to push different subjects into orthogonal attention subspaces, preventing feature interference while preserving individual identity characteristics. Extensive experiments demonstrate that MOSAIC achieves SOTA performance on multiple benchmarks. Notably, while existing methods typically degrade beyond 3 subjects, MOSAIC maintains high fidelity with **4+ reference subjects**, opening new possibilities for complex multi-subject synthesis applications.

## 1 Introduction

Multi-subject personalized generation in controllable image synthesis faces significant challenges in maintaining identity consistency while preventing attribute entanglement. Recent approaches have explored various strategies to address this problem: MS-Diffusion (Wang et al., 2025) and SSR-Encoder (Zhang et al., 2024c) incorporate spatial layout guidance into cross-attention layers to bind subjects to dedicated regions, while DreamO (Mou et al., 2025b) embeds routing constraints directly into DiT blocks for architectural-level control. TFCustom (Liu et al., 2025) takes a frequency-domain approach, introducing time-aware guidance to inject low- and high-frequency reference features at appropriate denoising stages. Most recently, XVerse (Chen et al., 2025) introduces token-specific modulation offsets for independent subject representation control.

However, these methods share a critical limitation: they lack explicit optimization of the underlying diffusion representations for both precise multi-subject alignment with target images and effective disentanglement between reference subjects. This shortcoming becomes increasingly severe as subject count grows, with most methods experiencing significant degradation beyond 3-4 subjects due

---

[*]Equal contribution.   [†]Project leader.   [‡]Corresponding author.
Source code available at: `https://github.com/bytedance-fanqie-ai/MOSAIC`

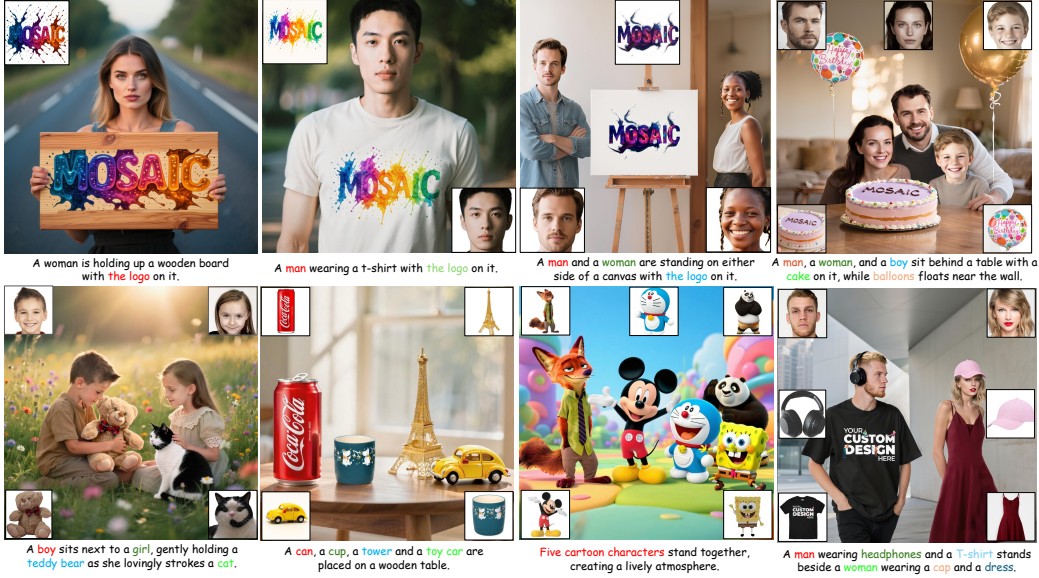

Figure 1: **MOSAIC** achieves high-quality personalized generation across single-subject and multi-subject scenarios. It demonstrates superior subject consistency in complex compositional settings.

to compounding feature interference. To understand why existing approaches fail, we analyze the fundamental requirements for multi-subject generation and identify two key deficiencies. The first concerns the correspondence problem: without explicit modeling of which specific reference image regions should attend to target latent parts, models cannot maintain semantic coherence across multiple subjects. The second involves multi-reference feature integration: when multiple subjects share the same latent space, their representations interfere, yet existing methods provide no mechanism to explicitly disentangle these conflated features. **These observations lead to a critical question: how can we design optimization objectives that simultaneously preserve individual subject fidelity while enforcing inter-subject separability between different subjects' representations?**

To address these fundamental limitations, we propose **MOSAIC**, a principled framework that reformulates multi-subject personalized generation as a representation optimization problem. The foundation of MOSAIC is the establishment of explicit semantic correspondences through a carefully curated dataset featuring densely annotated alignment points across reference-target image pairs. This correspondence foundation enables direct supervision of attention mechanisms for both reference-target alignment and inter-subject differentiation—a critical capability absent in existing approaches. Leveraging this semantic point correspondence, MOSAIC implements two complementary optimization objectives. The alignment objective employs cross-entropy supervision over attention distributions conditioned on correspondence labels, enforcing precise spatial mappings between reference and target representations. This mechanism ensures semantically coherent feature propagation from reference images to their designated target locations. Simultaneously, the disentanglement objective maximizes inter-subject attention divergence via symmetric KL regularization, promoting orthogonal attention patterns across reference subjects. This formulation effectively mitigates cross-subject feature interference—a persistent challenge in multi-reference scenarios. **Notably**, our method enables high-quality consistent generation with 4+ reference subjects, a capability that current approaches cannot achieve. Our contributions are as follows:

- We propose **MOSAIC**, a representation-centric framework that learns subject-consistent, disentangled representations by explicitly supervising attention correspondence between target and reference images, combining alignment and disentanglement objectives.

- We introduce **SemAlign-MS**, the first large-scale, annotated dataset with fine-grained semantic correspondences specifically tailored for multi-subject driven generation.

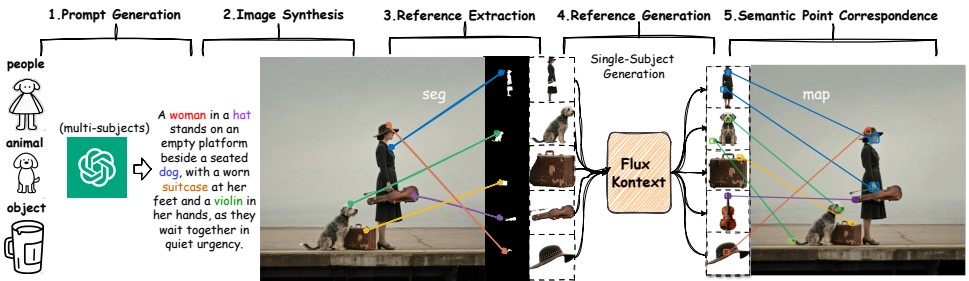

Figure 2: **SemAlign-MS Dataset Construction Pipeline.** Five-stage systematic pipeline for generating high-quality multi-reference training data with validated semantic correspondences.

- We demonstrate significant improvements over existing methods, particularly in challenging scenarios with 4+ subjects, while maintaining computational efficiency through our plug-and-play design.

## 2 RELATED WORKS

**Subject-Driven Image Generation.** Subject-driven image generation aims to synthesize images that preserve the identity and appearance of reference subjects while following textual descriptions. Recent advances have focused on improving reference encoding and attention mechanisms within diffusion frameworks. OminiControl (Tan et al., 2025) exploits the generative model itself as a reference image encoder, demonstrating the capability of diffusion transformers in maintaining subject consistency. For multi-subject scenarios, UNO (Wu et al., 2025b) proposes a systematic data generation pipeline, while DreamO (Mou et al., 2025a) constructs a router mechanism to focus attention on target subjects. XVerse (Chen et al., 2025) adopts text-stream modulation to transform reference images into token-specific offsets, enabling better integration of subject-specific cues. However, existing methods primarily rely on global feature matching, lacking explicit fine-grained detail constraints between reference and target regions. This limitation often results in imprecise spatial alignment, reduced reference fidelity, and attribute entanglement when handling multiple subjects simultaneously. Our work addresses these fundamental challenges through explicit semantic point correspondences that enable precise spatial alignment and cross-reference disentanglement, substantially enhancing subject consistency while preventing inter-subject interference.

**Visual Correspondence for Generation.** Visual correspondence establishes spatial relationships between semantically similar regions across images, serving as a foundation for various computer vision tasks (Lee et al., 2019; 2020; Lu et al., 2024). Traditional methods rely on handcrafted features like SIFT (Lowe, 2004) and SURF (Bay et al., 2006), while recent deep learning approaches leverage supervised learning with annotated datasets (Ham et al., 2017; Min et al., 2019; Yu et al., 2021). A promising direction has emerged using diffusion models for correspondence estimation. Methods such as DIFT (Tang et al., 2023), SD-DINO (Zhang et al., 2023), and GeoAware-SC (Zhang et al., 2024a) demonstrate that pre-trained diffusion features can establish reliable correspondences across diverse images without extensive supervision. However, these correspondences have not been effectively utilized for multi-subject generation tasks. Our work bridges this gap by being the first to leverage semantic point correspondences for multi-subject-driven generation. We establish a systematic pipeline that constructs high-quality correspondences and explicitly incorporates them into the generation process through our semantic corresponding attention alignment and multi-reference disentanglement mechanisms.

## 3 SEMALIGN-MS: A HIGH-QUALITY MULTI-SUBJECT DATASET WITH SEMANTIC POINT CORRESPONDENCES

Our data construction follows a systematic five-stage pipeline designed to ensure both diversity and quality. We first leverage GPT-4o (Hurst et al., 2024) with carefully designed templates to automati-

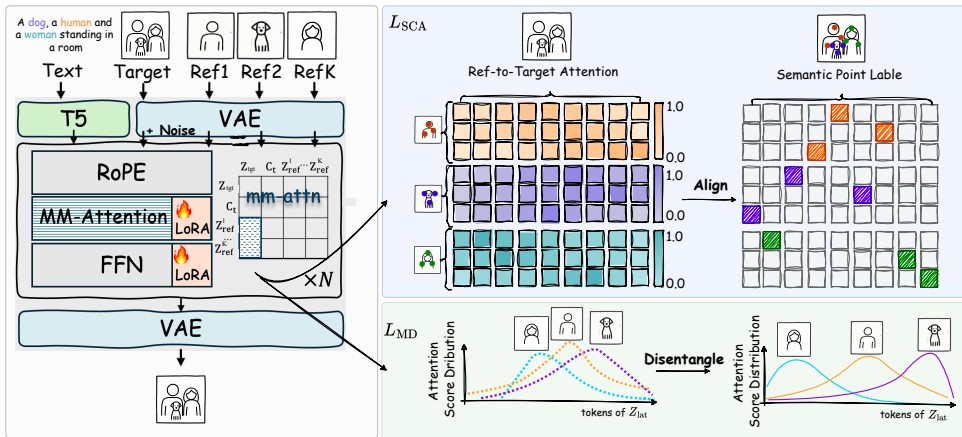

Figure 3: **Overview of MOSAIC Framework.** enables high-fidelity multi-subject generation through two complementary mechanisms. *(Blue region)* The **Semantic Correspondence Attention Loss (SCAL)** enforces precise point-to-point alignment between reference tokens and their corresponding locations in the target latent, ensuring high consistency. *(Green region)* The **Multi-Reference Disentanglement Loss (MDL)** maximizes the divergence between different references' attention distributions, pushing each subject into orthogonal representational subspaces.

cally generate diverse prompts containing multi-subject, encompassing various combinations of people, animals, objects, and their interactions to ensure comprehensive coverage of multi-subject scenarios commonly encountered in real-world applications. The generated prompts are then processed through state-of-the-art T2I models to synthesize images, where we implement a multi-criteria automated filtering strategy that evaluates image quality, subject clarity, and compositional coherence to retain only the highest-quality synthetic images. Subsequently, we employ Lang-SAM (Kirillov et al., 2023) for robust open-vocabulary detection and segmentation of all subjects within the synthesized images, enabling precise identification and isolation of individual subjects regardless of their semantic category and providing the foundation for subsequent correspondence establishment. Finally, we utilize FLUX Kontext (Labs et al., 2025) for viewpoint correction while maintaining semantic consistency, significantly enhancing dataset diversity and ensuring comprehensive appearance coverage across different viewpoints and poses.

Once we obtain the large-scale data, we establish *semantic point correspondences* between each target image and multiple reference images, where each reference image contributes $P^{(k)}$ sampled semantic points from the reference space that are mapped to corresponding locations in the target latent space, and $P^{(k)} \in \mathbb{N}$ is the number of semantic correspondence points for the $k$-th reference image. Formally, let $\mathcal{D} = \{(\{\mathcal{I}_{\text{ref}}^{(i,k)}\}_{k=1}^{K}, \mathcal{I}_{\text{tgt}}^{(i)})\}_{i=1}^{N}$ denote our dataset, where $N$ is the total number of target images in the training batch, $K$ is the maximum number of reference images, $\mathcal{I}_{\text{tgt}}^{(i)}$ represents the $i$-th target image, and $\mathcal{I}_{\text{ref}}^{(i,k)}$ denotes the $k$-th reference image for the $i$-th target. When fewer than $K$ reference images are available, we pad the remaining slots with black placeholder images to maintain consistent batch dimensions. For each pair $(\mathcal{I}_{\text{ref}}^{(i,k)}, \mathcal{I}_{\text{tgt}}^{(i)})$, we define the semantic correspondence set as:

$$\mathcal{C}^{(i,k)} = \{(u_{i,j}, v_{i,j})\}_{j=1}^{P^{(k)}} \tag{1}$$

where $u_{i,j}$ represents the $j$-th sampled point coordinate in reference image $\mathcal{I}_{\text{ref}}^{(k)}$, and $v_{i,j}$ denotes its corresponding semantic point in the target latent space. To prevent inter-reference conflicts where references map to the same target token, we enforce *correspondence disjointness* across different reference images:

$$\mathcal{V}^{(i,k)} \triangleq \{v_{i,j} \mid (u_{i,j}, v_{i,j}) \in \mathcal{C}^{(i,k)}\},$$
$$\text{s.t.} \quad \mathcal{V}^{(i,k_1)} \cap \mathcal{V}^{(i,k_2)} = \emptyset, \quad \forall k_1 \neq k_2, \tag{2}$$

where $\mathcal{V}^{(i,k)}$ represents the token position corresponding to the $j$-th sampled token of $k$-th reference image for the $i$-th training sample. This constraint ensures that each target latent location $v_{i,j}$ is associated with at most one reference image, preventing ambiguous supervision where multiple references compete for the same target region. Through this systematic pipeline, we successfully collect 1.2M high-quality image pairs with validated semantic correspondences that form the foundation of SemAlign-MS dataset. **Additional data pipeline details are provided in the Appendix.**

## 4 METHODOLOGY

### 4.1 OVERVIEW OF MOSAIC

**Architecture.** As shown in Fig. 3, given $K$ reference images $\{\mathbf{I}_{\text{ref}}^{(k)}\}_{k=1}^K$ depicting different subjects, a text prompt $\mathbf{T}$, and a target image $\mathbf{I}_{\text{tgt}}$ to be generated. The VAE encoder then transforms both the target image and all reference images into latent representations:

$$\mathbf{z}_{\text{tgt}} = \text{VAE}_{\text{enc}}(\mathbf{I}_{\text{tgt}}), \ \mathbf{z}_{\text{ref}}^{(k)} = \text{VAE}_{\text{enc}}(\mathbf{I}_{\text{ref}}^{(k)}) \tag{3}$$

where $k = 1, \ldots, K$. During training, noise is applied to the target image latent to obtain $\mathbf{z}_{\text{noise tgt}} = \mathbf{z}_{\text{tgt}} + \epsilon$, where $\epsilon \sim \mathcal{N}(0, 1)$. The text prompt is encoded using T5 (Raffel et al., 2023) to obtain text embeddings $\mathbf{C}_{\text{t}} = \text{T5}(\mathbf{T})$. To ensure spatial disentanglement between reference and target latents, modified Rotary Position Embeddings (RoPE) (Su et al., 2024) with distinct frequency bases are applied.

**Multi-Reference within MM-Attention.** The core of MOSAIC lies in how multiple reference latent and target latent interact through attention. Following OmniControl (Tan et al., 2025), we employ a LoRA-augmented branch for reference processing while maintaining the original model weights for the denoising branch. Crucially, we concatenate all reference latents into a unified representation to enable joint processing. In each transformer block $l$, the attention computation proceeds as:

$$\begin{aligned} \mathbf{Q}_{\text{tgt}}^{(l)}, \mathbf{K}_{\text{tgt}}^{(l)}, \mathbf{V}_{\text{tgt}}^{(l)} &= f_\theta(\mathbf{z}_{\text{noise tgt}}^{(l)}), \\ \mathbf{Q}_{\text{text}}^{(l)}, \mathbf{K}_{\text{text}}^{(l)}, \mathbf{V}_{\text{text}}^{(l)} &= f_\phi(\mathbf{C}_{\text{t}}), \end{aligned} \tag{4}$$

where $f_\theta$ and $f_\phi$ mean the mm-attention projection weights for target latent and text embedding. For the reference images, we first concatenate their latent representations:

$$\mathbf{z}_{\text{ref}}^{(l)} = [\mathbf{z}_{\text{ref}}^{(1,l)}; \mathbf{z}_{\text{ref}}^{(2,l)}; \ldots; \mathbf{z}_{\text{ref}}^{(K,l)}]. \tag{5}$$

Then process them through the LoRA branch of $f_\theta$, which denoted as $\Delta\theta_{\text{LoRA}}$:

$$\mathbf{Q}_{\text{ref}}^{(l)}, \mathbf{K}_{\text{ref}}^{(l)}, \mathbf{V}_{\text{ref}}^{(l)} = f_{\theta + \Delta\theta_{\text{LoRA}}}(\mathbf{z}_{\text{ref}}^{(l)}). \tag{6}$$

These features are then concatenated across modalities and processed through multi-head attention:

$$\begin{aligned} \mathbf{Q}^{(l)} &= [\mathbf{Q}_{\text{tgt}}^{(l)}; \mathbf{Q}_{\text{text}}^{(l)}; \mathbf{Q}_{\text{ref}}^{(l)}], \\ \mathbf{K}^{(l)} &= [\mathbf{K}_{\text{tgt}}^{(l)}; \mathbf{K}_{\text{text}}^{(l)}; \mathbf{K}_{\text{ref}}^{(l)}], \\ \mathbf{V}^{(l)} &= [\mathbf{V}_{\text{tgt}}^{(l)}; \mathbf{V}_{\text{text}}^{(l)}; \mathbf{V}_{\text{ref}}^{(l)}]. \end{aligned} \tag{7}$$

The attention scores are computed as $\mathbf{A}^{(l)} = \text{softmax}(\mathbf{Q}^{(l)}(\mathbf{K}^{(l)})^T / \sqrt{d})$, where $d$ is the feature dimension. Note that within $\mathbf{A}^{(l)}$, the reference-to-target latent attention sub-matrix $\mathbf{A}_{\text{ref} \to \text{tgt}}^{(l)} \in \mathbb{R}^{N_{\text{ref}} \times N_{\text{tgt}}}$ captures how each token from all reference images attends to the noisy latent, where $N_{\text{ref}}$ is the total number of tokens length across all reference images and $N_{\text{tgt}}$ is the token length of the target latent.

### 4.2 SEMANTIC CORRESPONDING ATTENTION ALIGNMENT

To faithfully preserve fine-grained semantic details, especially in regions requiring precise structural correspondence, we design the ***semantic correspondence attention loss (SCAL)*** to explicitly enforce point-wise semantic alignment within the reference-to-target attention mechanism.

As shown in Fig. 3, the reference-to-target latent attention allows each token from the reference images to attend to all token positions of the noised target latent. For a reference token at position $u$ and a target latent position $v$, the average reference to target latent attention across all **selected** DiT blocks $N_{\text{block}}$ is computed as:

$$\mathbf{A}_{\text{ref}\to\text{tgt}}[u,v] = \frac{1}{N_{\text{block}}} \sum_{l=1}^{N_{\text{block}}} \frac{\exp\left(\boldsymbol{Q}_u \boldsymbol{K}_v^\top / \sqrt{d}\right)}{\sum_{v=1}^{N_{\text{tgt}}} \exp\left(\boldsymbol{Q}_u \boldsymbol{K}_v^\top / \sqrt{d}\right)} \tag{8}$$

where $\boldsymbol{Q}_u$ and $\boldsymbol{K}_v$ are the query and key embeddings for the reference token at position $u$ and latent position $v$, respectively. To map local reference coordinates to global token indices in concatenated reference representation, we define:

$$\mathrm{G}(u_{i,j}^{(k)}) = \underbrace{\sum_{idx=1}^{k-1} N^{(idx)}}_{\text{offset}} + u_{i,j}^{(k)} \tag{9}$$

where $idx$ denotes the reference subject index. For each correspondence pair $(u_{i,j}^{(k)}, v_{i,j}^{(k)})$, to supervise the attention from the reference token at position $u_{i,j}^{(k)}$ that focused on its corresponding latent position $v_{i,j}^{(k)}$, we define:

$$\mathcal{L}_{\text{SCA}} = -\frac{1}{K}\sum_{k=1}^{K}\frac{1}{P^{(k)}}\sum_{j=1}^{P^{(k)}} \log \mathbf{A}_{\text{ref}\to\text{tgt}}[\mathrm{G}(u_{i,j}^{(k)}), v_{i,j}^{(k)}] \tag{10}$$

By integrating $\mathcal{L}_{\text{SCA}}$ into the training objective, our model is effectively encouraged to learn precise semantic mappings between reference and generated images. This leads to significantly improved preservation of local structure, textures, and fine details, going beyond the limitations of global similarity or implicit feature alignment.

### 4.3 MULTI-REFERENCE CORRESPONDING DISENTANGLEMENT

While alignment ensures high consistency, a crucial aspect of multi-subject driven generation is the potential interference among different reference images. We introduce ***multi-reference disentanglement loss (MDL)***, which promotes distinct attention patterns across references. MDL emphasizes the differentiation of attention maps from various subjects, thus preventing feature conflation.

Specifically, for $k$-th reference image, we collect the attention patterns at correspondence locations. For each correspondence point $(u_j, v_j) \in \mathcal{C}^{(i,k)}$:

$$\boldsymbol{a}_j^{(k)} = [\mathbf{A}_{\text{ref}\to\text{tgt}}[u_j, t] \mid t \in N_{tgt}] \in \mathbb{R}^{N_{tgt}} \tag{11}$$

Subsequently, we aggregate these attention responses for each reference:

$$\boldsymbol{a}^{(k)} = ||\frac{1}{P^{(k)}}\sum_{j=1}^{P^{(k)}}\boldsymbol{a}_j^{(k)} \in \mathbb{R}^{N_{tgt}}|| \tag{12}$$

where $||\cdot||$ denotes the normalization operation. Then the distance between $a^{(i)}$ and $a^{(j)}$ is:

$$\text{dist}(\boldsymbol{a}^{(i)}, \boldsymbol{a}^{(j)}) = \frac{1}{2}\mathcal{D}_{\text{KL}}(\boldsymbol{a}^{(i)}||\boldsymbol{a}^{(j)}) + \frac{1}{2}\mathcal{D}_{\text{KL}}(\boldsymbol{a}^{(j)}||\boldsymbol{a}^{(i)}) \tag{13}$$

where $\mathcal{D}_{\text{KL}}$ is KL divergence. Subsequently, we enforce distinct attention patterns across references:

$$\mathcal{L}_{\text{MD}} = -\frac{1}{K(K-1)}\sum_{i=1}^{K}\sum_{j=1,j\neq i}^{K} \text{dist}(\boldsymbol{a}^{(i)}, \boldsymbol{a}^{(j)}) \tag{14}$$

By maximizing attention pattern divergence, this loss prevents references from competing for the same attention regions, mitigating cross-reference feature interference. The overall loss is defined as:

$$\mathcal{L} = \mathcal{L}_{\text{diff}} + \alpha \mathcal{L}_{\text{SCA}} + \beta \mathcal{L}_{\text{MD}} \tag{15}$$

where $\mathcal{L}_{\text{diff}}$ is the flow-matching loss used in (Esser et al., 2024), $\alpha$ and $\beta$ are two factors that balance the weight.

Table 1: Quantitative comparison for single-subject and multi-subject on DreamBench benchmark.

| Reference | Method | CLIP-I ↑ | CLIP-T ↑ | DINO ↑ | UnifiedReward ↑ | HPSv3 ↑ |
|---|---|---|---|---|---|---|
| Single-Subject | DreamBooth | 80.30 | 30.52 | 66.81 | 3.30 | 8.40 |
| | BLIP-Diffusion | 80.47 | 30.24 | 69.82 | 3.05 | 8.96 |
| | SSR-Encoder | 82.10 | 30.79 | 61.22 | 3.27 | 9.25 |
| | MS-Diffusion | 80.82 | 31.05 | 70.32 | 3.70 | 9.40 |
| | UNO | 83.50 | 30.41 | 75.97 | 4.00 | 11.24 |
| | DreamO | 83.35 | 30.61 | 76.03 | 4.33 | 12.78 |
| | XVerse | 83.20 | 30.20 | 75.44 | 4.20 | 11.02 |
| | **MOSAIC** | **84.30** | **31.64** | **77.40** | **4.40** | **14.36** |
| Multi-Subject | MS-Diffusion | 72.60 | 31.91 | 52.50 | 3.67 | 8.25 |
| | UNO | 73.29 | 32.23 | 54.22 | 4.23 | 11.55 |
| | DreamO | 73.32 | 32.10 | 52.17 | 4.33 | 13.25 |
| | XVerse | 73.47 | 31.20 | 53.71 | 3.87 | 11.10 |
| | **MOSAIC** | **76.30** | **32.40** | **56.83** | **4.39** | **14.90** |

Table 2: Quantitative results of single-subject and multi-subject driven generation on XVerseBench.

| Method | Single-Subject | | | | | Multi-Subject | | | | | Overall |
|---|---|---|---|---|---|---|---|---|---|---|---|
| | DPG | ID-Sim | IP-Sim | AES | AVG | DPG | ID-Sim | IP-Sim | AES | AVG | |
| MS-Diffusion | 96.89 | 6.52 | 55.71 | 59.63 | 54.69 | 87.21 | 3.77 | 46.21 | **55.91** | 48.28 | 51.49 |
| UNO | 89.65 | 47.91 | 80.40 | 55.90 | 68.47 | 85.28 | 31.82 | 67.00 | 54.24 | 59.59 | 64.03 |
| DreamO | **96.93** | 75.48 | 70.84 | 54.57 | 74.46 | 88.80 | 50.24 | 64.63 | 52.47 | 64.04 | 69.25 |
| OmniGen2 | 92.60 | 62.41 | 74.08 | 52.34 | 70.36 | **91.55** | 40.81 | 67.15 | 51.40 | 62.73 | 66.55 |
| XVerse | 93.69 | 79.48 | 76.86 | 56.84 | 76.72 | 88.26 | 66.59 | 71.48 | 53.97 | 70.08 | 73.40 |
| **MOSAIC** | 96.55 | **81.98** | **80.92** | **60.77** | **80.05** | 88.94 | **69.90** | **74.27** | 55.02 | **72.03** | **76.04** |

## 5 EXPERIMENTS

### 5.1 EXPERIMENTAL DETAILS

**Implementation Details**   In this work, we adopt FLUX-1.0-DEV (Labs, 2024) as our base model. The rank of additional LoRA (Hu et al., 2022) is set as 128. During training, we employ the AdamW (Kingma & Ba, 2015) optimizer with a learning rate of 1e-4 and train the model for 100K steps, using a batch size of 1 per GPU. The $\alpha$ and $\beta$ in Eq. 15 are set to 0.4 and 0.6.

**Evaluation Setting**   We evaluate our proposed method on DreamBench (Ruiz et al., 2023) and XVerseBench (Chen et al., 2025), focusing on both single-subject and multi-subject generation scenarios. For comparison, we include DreamBooth (Ruiz et al., 2023), BLIP-Diffusion (Li et al., 2023), SSR-Encoder (Zhang et al., 2024b), MS-Diffusion (Wang et al., 2025), UNO (Wu et al., 2025b), DreamO (Mou et al., 2025a), OmniGen2 (Wu et al., 2025a), and XVerse (Chen et al., 2025). DreamBench utilizes the CLIP-I (Radford et al., 2021), DINO (Oquab et al., 2023), and CLIP-T (Radford et al., 2021) to evaluate the semantic and text-image alignment. XVerseBench utilizes DPG score (Hu et al., 2024), ID-Sim (Deng et al., 2019), IP-Sim (Oquab et al., 2023), AES (discus0434, 2024) to further evaluate the alignment and image quality.

### 5.2 MAIN RESULTS

**Quantitative Results**   The results on DreamBench and XVerseBench are shown in Tab.1 and Tab.2, respectively.

On DreamBench, MOSAIC achieves consistently strong results across all metrics. In the single-subject setting, it achieves 84.30 (CLIP-I), 31.64 (CLIP-T), and 77.40 (DINO), surpassing the second-best method UNO by 0.80 on CLIP-I, and the second-best method DreamO by 1.37 on DINO, indicating better image-text alignment and semantic fidelity.In the more challenging multi-subject setting, it maintains strong performance with 76.30 (CLIP-I), 32.40 (CLIP-T), and 56.83 (DINO), outperforming second-best competitors by approximately 3.0, 0.2, and 2.6 points, respectively. Furthermore, MOSAIC demonstrates a significant advantage in VLM-based metrics across both scenarios, which align more closely with human visual preferences. It achieves the highest

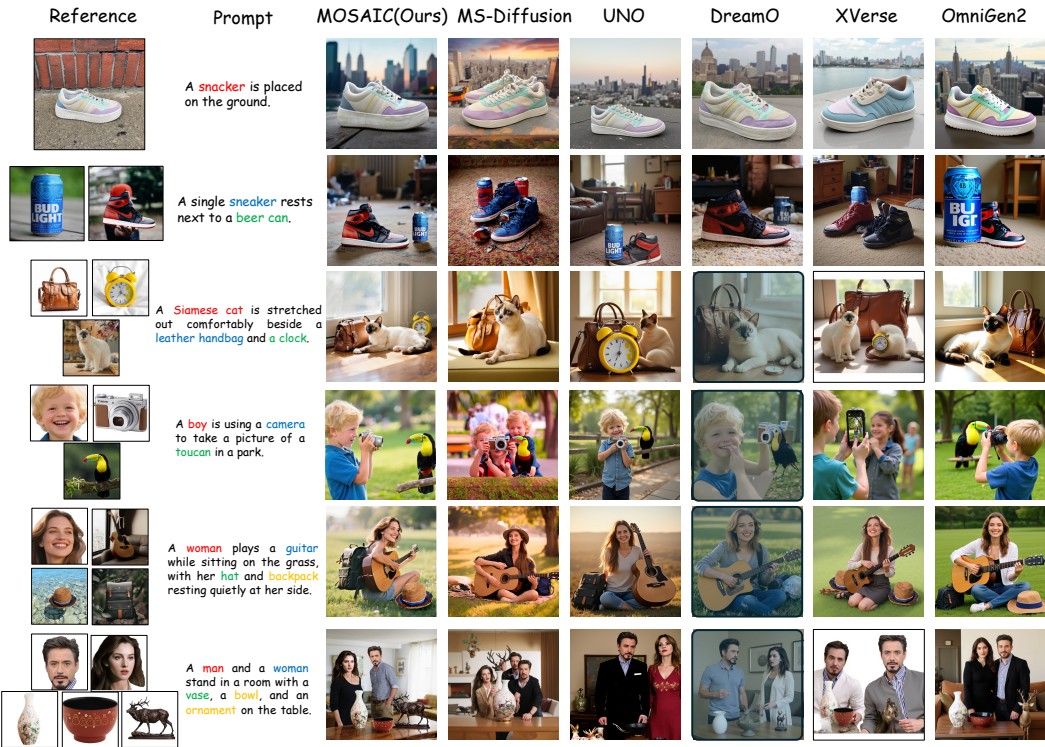

Figure 4: Qualitative comparison on single and multi-subject driven generation. DreamO does not support 3+ reference images; masked regions show where comparison is unavailable, with visible results generated from two randomly selected references.

Table 3: Quantitative comparison with strong multi-subject/identity baselines. We report Identity Preservation (IP) and Text Consistency (PC) scores under zero-shot settings. MOSAIC significantly outperforms FastComposer and Face-Diffuser, particularly in multi-subject identity preservation.

| Methods | Single-Subject | | Multi-Subject | |
|---|---|---|---|---|
| | IP ↑ | PC ↑ | IP ↑ | PC ↑ |
| FastComposer (zero-shot) | 0.514 | 0.243 | 0.465 | 0.233 |
| Subject-Diffusion (zero-shot) | 0.605 | 0.228 | 0.435 | 0.210 |
| Face-Diffuser (zero-shot) | 0.708 | 0.325 | 0.594 | 0.320 |
| **MOSAIC (zero-shot)** | **0.785** | **0.362** | **0.712** | **0.358** |

scores in UnifiedReward (**4.40** for single-subject and **4.39** for multi-subject) and HPSv3 (**14.36** and **14.90** ), substantially outperforming the state-of-the-art method DreamO. These results strongly validate that MOSAIC not only excels in traditional metrics but also possesses superior visual aesthetic quality and comprehensive image-text consistency regardless of the subject complexity.

On XVerseBench, MOSAIC achieves the highest overall average score of 76.04, surpassing XVerse's 73.40. And MOSAIC achieves a DPG score that is on par with the top-performing DreamO (96.93). It also demonstrates clear advantages in identity preservation, with ID-Sim scores of 81.98 for single-object and 69.90 for multi-object scenarios, as well as in perceptual similarity, with IP-Sim scores of 80.92 and 74.27, respectively. Overall, these demonstrate the robustness of our proposed method under varying subject complexity.

**Comparison with Strong Identity-Preserving Methods** To further validate MOSAIC's effectiveness against strong multi-subject and identity-preserving methods, we conducted supplementary comparisons following the official evaluation protocol of Face-Diffuser (Wang et al., 2024). We compared our method against FastComposer (Xiao et al., 2023), Subject-Diffusion (Ma et al., 2024),

Table 4: Impacts of different losses on DreamBench benchmark in multi-subject scenario.

| $\mathcal{L}_{\text{SCA}}$ | $\mathcal{L}_{\text{MD}}$ | CLIP-I | CLIP-T | DINO |
|:---:|:---:|:---:|:---:|:---:|
| ✗ | ✗ | 73.45 | 29.90 | 52.03 |
| ✓ | ✗ | 75.89 | 31.10 | 55.99 |
| ✗ | ✓ | 75.10 | 31.70 | 55.24 |
| ✓ | ✓ | 76.30 | 32.40 | 56.83 |

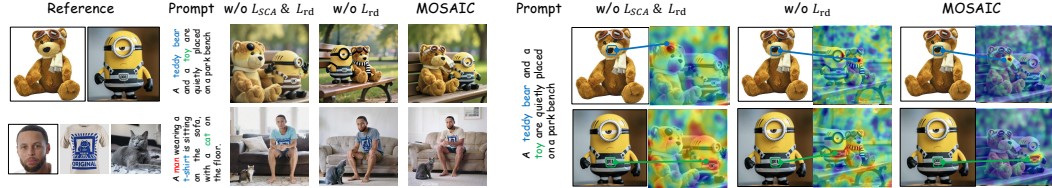

Figure 5: (Left): Ablation study of MOSAIC. (Right): Visualization of attention maps from specific reference regions to generated image regions. **Zoom in for details.**

and Face-Diffuser (Wang et al., 2024) using Identity Preservation (IP) and Text Consistency (PC) metrics under both single-subject and multi-subject settings.

The results are presented in Tab. 3. MOSAIC exhibits significant advantages over these strong baselines. In the **Single-Subject Scenario**, our method achieves an IP score of **0.785**, substantially outperforming Face-Diffuser (0.708) and FastComposer (0.514). The advantage is even more pronounced in the **Multi-Subject Scenario**, where MOSAIC achieves an IP score of **0.712**, far surpassing Face-Diffuser's 0.594. We attribute this superior performance to our proposed fine-grained alignment ($\mathcal{L}_{\text{SCA}}$) and spatial decoupling mechanism ($\mathcal{L}_{\text{MD}}$), which effectively resolve feature confusion between multiple subjects, thereby achieving state-of-the-art levels in both identity preservation and text consistency.

**Qualitative Results** Fig. 4 presents qualitative results across scenes with varying numbers of reference objects, demonstrating MOSAIC's superior performance in multi-subject generation. **Appearance consistency.** Our method maintains strong visual coherence across different subjects. For instance, the snackers in rows 1-2 and the can in row 2 preserve consistent textures and shapes throughout the generated scenes, indicating effective feature transfer from reference images. **Multi-subject handling.** The advantages of our approach become particularly evident with three or more reference subjects. In row 3, competing methods suffer from object omission and duplication: MS-Diffusion and OmniGen2 fail to render the clock entirely, while XVerse produces duplicated and deformed cats. In contrast, MOSAIC accurately represents all three objects without such artifacts. **Scalability to more complex scenes.** For scenes involving four or more reference objects, existing methods exhibit substantial degradation. As shown in row 6, only MOSAIC preserves facial consistency across multiple subjects, while other approaches show significant quality loss and identity confusion. Overall, these results highlight MOSAIC's ability to generate high-quality, consistent outputs across diverse compositional scenarios, especially in challenging multi-subject settings where other methods fall short.

## 5.3 Ablation Studies

**Impact of Different Losses** We conduct ablation studies to evaluate each component of our method on multi-subjects of Dreambench, as shown in Tab.4. Adding the semantic correspondence attention loss $\mathcal{L}_{SCA}$ leads to consistent gains across all metrics, with CLIP-I improving from 73.45 to 75.89, CLIP-T from 29.90 to 31.10, and DINO from 52.03 to 55.99. This confirms that point-wise semantic supervision aids fine-grained detail retention. Conversely, adding $L_{MD}$ contributes more significantly to text alignment (CLIP-T), improving it by +1.80 (compared to +1.20 from $L_{SCA}$). Introducing $\mathcal{L}_{SCA}$ and $\mathcal{L}_{MD}$ further improves CLIP-I to 76.30, CLIP-T to 32.40, and DINO to 56.83, suggesting that suppressing cross-reference feature conflation strengthens semantic fidelity and compositional coherence. Moreover, visual results in Fig. 5(Left) improve progressively: the

baseline yields blurry, inconsistent outputs, while adding $\mathcal{L}_{\text{SCA}}$ enhances composition and detail. The full model achieves both global scene coherence and local identity preservation by preventing semantic interference across references.

**Visualization of attention maps** Fig. 5(Right) visualizes attention maps between reference features and target latents, revealing how our proposed losses improve spatial correspondence. We examine specific reference regions (e.g., teddy bear's nose, numerical digits on toys) and visualize their attention distributions across the target image, with attention weights averaged across transformer layers. The baseline model shows dispersed and misaligned attention patterns, where reference tokens often activate incorrect target regions, leading to feature mixing and loss of subject identity. Introducing $\mathcal{L}_{\text{SCA}}$ guides reference tokens toward semantically appropriate target locations, yet the attention remains broadly distributed with notable overlap between different subjects (e.g., bear features bleeding into Minion regions). The addition of $\mathcal{L}_{\text{MD}}$ significantly sharpens the attention focus—each reference token now precisely attends to its corresponding target region while suppressing responses to other subjects' areas. This progression (baseline $\rightarrow \mathcal{L}_{\text{SCA}} \rightarrow \mathcal{L}_{\text{SCA}} + \mathcal{L}_{\text{MD}}$) confirms our hypothesis: alignment loss establishes semantic correspondence while disentanglement loss enforces spatial exclusivity, together enabling accurate multi-subject synthesis.

**Mechanism Analysis: Step-by-Step Denoising.** To intuitively understand how our proposed modules influence the generation process, we visualize the step-by-step denoising evolution (see Figure 7 in Appendix). Without our specific losses, the baseline model struggles with fine-grained identity preservation, often generating generic objects that lack specific design details found in the reference images. Introducing $\mathcal{L}_{SCA}$ successfully anchors these identity features; however, without the disentanglement constraint ($\mathcal{L}_{MD}$), we observe severe feature entanglement. For instance, in multi-subject scenarios, visual attributes from one subject (e.g., the distinct yellow color of a toy car) frequently bleed onto adjacent subjects (e.g., tinting a dog's fur with an unnatural yellowish hue). By further integrating $\mathcal{L}_{MD}$, this feature confusion is effectively suppressed. The model spatially decouples the attributes of different subjects, restoring their correct appearances while keeping distinctive features strictly confined to their respective regions. This confirms that $\mathcal{L}_{SCA}$ is responsible for identity anchoring, while $\mathcal{L}_{MD}$ is indispensable for enforcing spatial exclusivity and preventing cross-subject attribute leakage.

## 6 CONCLUSION

In this paper, we propose MOSAIC, a representation-centric approach for multi-subject driven image generation. Our method features a semantic correspondence pipeline, along with explicit attention alignment and disentanglement mechanisms, effectively addressing the key challenges of identity preservation and attribute entanglement in multi-subject personalized generation scenarios. We also curate and will release SemAlign-MS, a large-scale multi-subject dataset with fine-grained semantic point correspondences to facilitate future research in controllable generation.

Extensive experiments demonstrate that MOSAIC achieves state-of-the-art performance in both identity fidelity and semantic consistency across established benchmarks. **A key strength of our method is its superior fidelity-diversity balance: unlike global constraints, our point-wise alignment targets only critical semantic features, preserving substantial freedom for non-essential attributes like pose and lighting. This enables both high generation diversity and identity preservation.** Detailed analysis is provided in **Appendix A.4**. Furthermore, our approach exhibits robust scalability, successfully handling complex compositions with **4+** reference subjects.

**Limitations.** While MOSAIC achieves strong multi-subject generation, it comes with increased computational cost during training. **Future work will explore efficient multi-reference encoding strategies and attention compression techniques to reduce computational overhead while maintaining generation quality.**

**Use of LLMs.** We utilize LLMs to assist with formula derivations and writing refinement.

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

# A  ADDITIONAL QUALITATIVE EVALUATION AND MECHANISM ABLATION ANALYSIS

## A.1  ABLATION ANALYSIS

### A.1.1  IMPACT OF THE SEMALIGN-MS DATASET

Table 5: The impact of the SEMALIGN-MS dataset. $*$ means trained ont the SEMALIGN-MS dataset but has no SCAL and MDL.

| Method | Single-Subject | | | | | Multi-Subject | | | | | Overall |
|---|---|---|---|---|---|---|---|---|---|---|---|
| | DPG | ID-Sim | IP-Sim | AES | AVG | DPG | ID-Sim | IP-Sim | AES | AVG | |
| MS-Diffusion | 96.89 | 6.52 | 55.71 | 59.63 | 54.69 | 87.21 | 3.77 | 46.21 | 55.91 | 48.28 | 51.49 |
| UNO | 89.65 | 47.91 | 80.40 | 55.90 | 68.47 | 85.28 | 31.82 | 67.00 | 54.24 | 59.59 | 64.03 |
| DreamO | 96.93 | 75.48 | 70.84 | 54.57 | 74.46 | 88.80 | 50.24 | 64.63 | 52.47 | 64.04 | 69.25 |
| OmniGen2 | 92.60 | 62.41 | 74.08 | 52.34 | 70.36 | 91.55 | 40.81 | 67.15 | 51.40 | 62.73 | 66.55 |
| XVerse | 93.69 | 79.48 | 76.86 | 56.84 | 76.72 | 88.26 | 66.59 | 71.48 | 53.97 | 70.08 | 73.40 |
| **MOSAIC**$^{*}$ | 91.24 | 75.11 | 75.62 | 56.37 | 74.59 | 85.46 | 60.09 | 68.55 | 53.39 | 66.87 | 70.73 |
| **MOSAIC** | 96.55 | **81.98** | **80.92** | **60.77** | **80.05** | 88.94 | **69.90** | **74.27** | 55.02 | **72.03** | **76.04** |

To further substantiate the effectiveness of the proposed SCAL and MDL modules, we performed ablation experiments to disentangle improvements introduced by the dataset itself. The results on XVerseBench are presented in Tab. 5. Here, MOSAIC* denotes the baseline trained on the SEMALIGN-MS dataset without SCAL or MDL. As illustrated, integrating SCAL and MDL leads to consistent gains across both single-reference and multi-reference settings. Specifically, ID-Sim improves by 6.87% and 9.81%, while IP-Sim increases by 5.30% and 5.72%, respectively. Moreover, the overall score rises substantially from 70.73% to 76.04%, highlighting the effectiveness of our approach.

### A.1.2  COMPONENT ANALYSIS OF SCAL AND MDL

To provide a deeper understanding of the individual contributions of our proposed modules, we conducted a comprehensive ablation study on the DreamBench multi-reference scenario. We evaluated the model performance under settings containing only $L_{SCA}$, only $L_{MD}$, and the full combination. The results are summarized in Table 6.

Table 6: Ablation study of SCAL and MDL modules on DreamBench multi-reference scenario. The results indicate that $L_{SCA}$ dominates visual identity preservation (CLIP-I, DINO), while $L_{MD}$ contributes significantly to text alignment (CLIP-T) through spatial disentanglement. The full MOSAIC achieves the best performance via their synergy.

| $L_{SCA}$ | $L_{MD}$ | CLIP-I | CLIP-T | DINO |
|---|---|---|---|---|
| × | × | 73.45 | 29.90 | 52.03 |
| ✓ | × | 75.89 | 31.10 | 55.99 |
| × | ✓ | 75.10 | 31.70 | 55.24 |
| ✓ | ✓ | **76.30** | **32.40** | **56.83** |

Based on these results, we make the following observations:

$L_{SCA}$ **Dominates Visual Identity Preservation.** Comparing the single-module results, $L_{SCA}$ yields a more significant improvement in visual metrics (CLIP-I +2.44, DINO +3.96) compared to $L_{MD}$. This is because $L_{SCA}$ enforces strict subject-level attention alignment between the reference and generated images. This strong visual constraint directly promotes the retention of fine-grained identity features.

$L_{MD}$ **Prioritizes Text Alignment & Disentanglement.** Conversely, $L_{MD}$ contributes more significantly to Text Alignment (CLIP-T), improving it by +1.80 (compared to +1.20 from $L_{SCA}$). The core function of $L_{MD}$ is to explicitly decouple multiple subjects within the prompt (e.g., distin-

guishing "a dog" from "a cat"). Without this proper spatial decoupling, the model is prone to subject confusion, which lowers prompt adherence.

**Synergistic Effect.** When combined, the model achieves the best performance across all metrics. This confirms that strong visual feature alignment ($L_{SCA}$) and spatial multi-subject decoupling ($L_{MD}$) are complementary and essential for high-quality multi-subject generation.

### A.1.3 GENERALIZATION TO EXTERNAL DATASETS

To demonstrate that MOSAIC is not limited to our SemAlign-MS dataset and can generalize to other data sources, we conducted supplementary experiments on two widely used public datasets: Subject-200K Tan et al. (2025) (Single-Subject) and X2I-subject-driven Xiao et al. (2024) (Multi-Reference). We applied our automated pipeline to extract semantic points for these datasets without manual intervention.

**Performance on Subject-200K.** Since this dataset contains only single subjects, we enabled only $L_{SCA}$. As shown in Table 7, our method achieves significant improvements over the baseline on DreamBench single-reference metrics.

Table 7: Generalization performance on Subject-200K dataset (Single-Subject Scenario).

| $L_{SCA}$ | $L_{MD}$ | CLIP-I | CLIP-T | DINO |
|:---:|:---:|:---:|:---:|:---:|
| × | N/A | 81.25 | 30.05 | 74.60 |
| ✓ | N/A | **83.32** | **30.21** | **75.57** |

**Performance on X2I-subject-driven.** For this multi-subject dataset, we enabled both losses. The results in Table 8 confirm that MOSAIC effectively generalizes to external multi-subject data, significantly boosting CLIP-I and DINO scores while maintaining high text alignment.

Table 8: Generalization performance on X2I-subject-driven dataset (Multi-Subject Scenario).

| $L_{SCA}$ | $L_{MD}$ | CLIP-I | CLIP-T | DINO |
|:---:|:---:|:---:|:---:|:---:|
| × | × | 71.30 | 29.65 | 51.73 |
| ✓ | × | 75.35 | 31.08 | 55.49 |
| × | ✓ | 74.92 | 31.60 | 55.06 |
| ✓ | ✓ | **75.70** | **32.15** | **56.20** |

These results validate that leveraging our high-accuracy automated pipeline, MOSAIC can be easily transferred to other datasets to consistently enhance visual consistency and text control.

### A.1.4 GENERALIZATION TO DIFFERENT BACKBONES (SDXL)

A key concern raised regarding the fairness of comparison is whether MOSAIC's performance gains are solely due to the powerful FLUX backbone. To demonstrate that our proposed modules ($L_{SCA}$ and $L_{MD}$) are architecture-agnostic and generalize well to other backbones (e.g., from DiT to UNet), we applied MOSAIC to the widely used Stable Diffusion XL (SDXL) model.

We compared our "MOSAIC-SDXL" variant against mainstream SDXL-based personalization methods on the DreamBench benchmark. As shown in Table 9, MOSAIC-SDXL consistently outperforms existing baselines in both single-subject and multi-subject settings.

These results confirm that the effectiveness of SCAL and MDL is not dependent on a specific backbone architecture but represents a robust contribution to the general personalization framework.

### A.1.5 IMPACT OF SAMPLED NUMBER $P^{(k)}$

As shown in Fig. 6, increasing $P^{(k)}$ from 50 to 200 consistently improves performance: CLIP-I rises from 73.9 to 76.4, DINO from 53.8 to 57.0, and CLIP-T from 30.1 to 32.6. These gains indicate that denser semantic correspondences enhance attention supervision and fine-grained spatial alignment.

Table 9: Generalization performance on SDXL backbone. We compare MOSAIC-SDXL against other SDXL-based methods on DreamBench. The results demonstrate that our method maintains SOTA performance even when applied to a UNet-based architecture, validating the robustness of our loss designs.

| Reference | Method | CLIP-I ↑ | CLIP-T ↑ | DINO ↑ |
|---|---|---|---|---|
| Single-Subject | DreamBooth | 80.30 | 30.52 | 66.81 |
| | BLIP-Diffusion | 80.47 | 30.24 | 69.82 |
| | SSR-Encoder | 82.10 | 30.79 | 61.22 |
| | MS-Diffusion | 80.82 | 31.05 | 70.32 |
| | **MOSAIC-SDXL (Ours)** | **82.15** | **31.20** | **72.45** |
| Multi-Subject | MS-Diffusion | 72.60 | 31.91 | 52.50 |
| | **MOSAIC-SDXL (Ours)** | **73.20** | **32.11** | **53.84** |

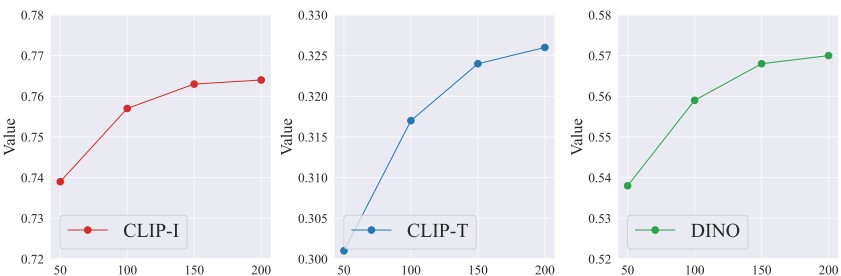

Figure 6: Impact of different $P^{(k)}$ on DreamBench benchmark in multi-subject scenario.

However, performance gains diminish beyond a certain point, with increased computational cost. We thus set $P^{(k)} = 150$ by default to balance accuracy and efficiency.

### A.1.6   IMPACT OF APPLYING SCAL AND MDL TO DIFFERENT ATTENTION BLOCKS

Table 10: The impact of the number of attention blocks applied SCAL and MDL. $e$, $m$, and $l$ denote applying SCAL and MDL primarily on early, middle, and late attention blocks, respectively.

| Method | Single-Subject | | | | | Multi-Subject | | | | | Overall |
|---|---|---|---|---|---|---|---|---|---|---|---|
| | DPG | ID-Sim | IP-Sim | AES | AVG | DPG | ID-Sim | IP-Sim | AES | AVG | |
| **MOSAIC\*** | 91.24 | 75.11 | 75.62 | 56.37 | 74.59 | 85.46 | 60.09 | 68.55 | 53.39 | 66.87 | 70.73 |
| **MOSAIC$^e$** | 96.55 | 81.98 | 80.92 | 60.77 | 80.05 | 88.94 | 69.90 | 74.27 | 55.02 | 72.03 | 76.04 |
| **MOSAIC$^{e+m}$** | 94.87 | 79.42 | 78.36 | 58.13 | 77.70 | 87.25 | 65.71 | 71.92 | 54.21 | 69.77 | 73.74 |
| **MOSAIC$^{e+m+l}$** | 93.02 | 76.84 | 77.11 | 57.42 | 76.10 | 86.27 | 62.35 | 70.48 | 53.92 | 68.26 | 72.18 |

We investigate the optimal placement of our proposed SCAL and MDL modules across different attention layers. Table 10 presents ablation results comparing various layer configurations against our baseline model (MOSAIC*).

**Overall effectiveness validation.** Regardless of placement, all configurations consistently outperform the baseline, demonstrating the robustness and universal applicability of our SCAL and MDL design. This consistent improvement across different architectural positions validates the fundamental effectiveness of our approach.

**Early layers achieve optimal performance.** The configuration applying SCAL and MDL primarily to early attention blocks (MOSAIC$^e$) delivers the best results, achieving an overall score of 76.04 across both single- and multi-subject scenarios. This superior performance stems from the architectural characteristics of early attention layers, which primarily handle global structure and semantic alignment. At this stage, reference-to-target attention can establish more accurate positional correspondences, facilitating robust cross-image semantic consistency and effective subject separation.

**Diminishing gains in deeper layers.** Extending supervision to middle and late attention blocks (MOSAIC[e+m] and MOSAIC[e+m+l]) yields progressively smaller gains. This performance plateau occurs because deeper layers increasingly focus on local details rather than global correspondences. The emphasis on fine-grained features in these layers makes global position mappings between reference and target images less relevant. Consequently, applying our constraints at deeper layers creates a trade-off between detail enhancement and semantic consistency preservation, partially offsetting the benefits of our proposed losses.

**Implications for design choices.** These findings reveal that early attention layers provide the optimal insertion point for SCAL and MDL, effectively balancing semantic alignment with computational efficiency. This configuration maximizes improvements across all evaluation metrics: text–image consistency (DPG), identity preservation (ID-Sim), subject disentanglement (IP-Sim), and overall aesthetic quality (AES).

### A.1.7    COMPARISON WITH STRONG MULTI-SUBJECT BASELINES

To ensure a comprehensive evaluation against state-of-the-art identity-preserving methods, we conducted supplementary comparisons against strong baselines including FastComposer Xiao et al. (2023), Subject-Diffusion Ma et al. (2024), and Face-Diffuser Wang et al. (2024). Following the official evaluation settings of Face-Diffuser, we evaluated Identity Preservation (IP) and Text Consistency (PC) under both Single-Subject and Multi-Subject settings.

The quantitative results are presented in Table 11. MOSAIC exhibits significant advantages over these methods:

- **Single-Subject Scenario:** Our method achieves an IP score of **0.785**, substantially outperforming Face-Diffuser (0.708) and FastComposer (0.514).

- **Multi-Subject Scenario:** The advantage is even more pronounced. MOSAIC achieves an IP score of **0.712**, far surpassing Face-Diffuser's 0.594.

We attribute this superior performance to our fine-grained alignment ($L_{SCA}$) and spatial decoupling mechanism ($L_{MD}$), which effectively resolve feature confusion between multiple subjects, thereby achieving state-of-the-art levels in both identity preservation and text consistency.

Table 11: Quantitative comparison with strong multi-subject baselines. We report Identity Preservation (IP) and Text Consistency (PC) scores under zero-shot settings. MOSAIC significantly outperforms FastComposer and Face-Diffuser, particularly in multi-subject identity preservation.

| Methods | Single-Subject | | Multi-Subject | |
|---|---|---|---|---|
| | IP ↑ | PC ↑ | IP ↑ | PC ↑ |
| FastComposer (zero-shot) | 0.514 | 0.243 | 0.465 | 0.233 |
| Subject-Diffusion (zero-shot) | 0.605 | 0.228 | 0.435 | 0.210 |
| Face-Diffuser (zero-shot) | 0.708 | 0.325 | 0.594 | 0.320 |
| **MOSAIC (zero-shot)** | **0.785** | **0.362** | **0.712** | **0.358** |

### A.1.8    IMPACT OF LoRA RANK

To determine the optimal model capacity for capturing fine-grained semantic correspondences, we conducted a detailed ablation study on the LoRA rank setting, evaluating performance on the Dream-Bench multi-reference scenario with ranks set to 32, 64, 128, and 256. As presented in Table 12, we observe two distinct phases regarding performance scaling. Initially, a Performance Growth phase is evident from Rank 32 to 128, where increasing the rank expands the learnable parameter capacity, leading to a significant upward trend across all metrics (CLIP-I, CLIP-T, DINO). This indicates that a sufficient parameter budget is essential for the model to learn complex spatial decoupling and precise point-wise alignment. However, this is followed by a Performance Saturation phase from Rank 128 to 256, where performance gains become negligible (e.g., CLIP-I improves by only 0.03), exhibiting clear diminishing returns. Consequently, considering the trade-off between performance gains and training costs (VRAM usage and computational load), we selected **Rank = 128** as the

Table 12: Ablation study on LoRA rank. We compare performance across different rank settings (32, 64, 128, 256) on DreamBench. Rank 128 achieves the best balance between performance and parameter efficiency, while Rank 256 offers only marginal improvements.

| LoRA Rank | CLIP-I | CLIP-T | DINO |
|---|---|---|---|
| 32 | 74.30 | 31.55 | 54.02 |
| 64 | 75.71 | 32.10 | 55.90 |
| **128** | **76.30** | **32.40** | **56.83** |
| 256 | 76.33 | 32.45 | 56.85 |

optimal default setting, as it ensures sufficient capacity for high-quality alignment while avoiding unnecessary resource overhead.

Table 13: Inference speed comparison of consistency-based models on NVIDIA H20 GPU at 512×512 resolution. MOSAIC achieves competitive performance while supporting multiple reference inputs.

| **Method** | UNO | DreamO | XVerse | **MOSAIC** |
|---|---|---|---|---|
| **Inference Speed (s/it)** | 2.05 | 1.16 | 1.87 | 1.53 |

Table 14: Computational cost analysis of SCAL with different sampling points. We measure the training time overhead per training step. $P^{(k)} = 150$ represents the optimal balance, offering significant performance gains with acceptable computational cost, whereas $P^{(k)} = 200$ introduces excessive overhead.

| Sampled Points | Training Time per Step (s) | Inference Time per Step (s) |
|---|---|---|
| 0 | 2.85 | 1.53 |
| 50 | 3.10 | 1.53 |
| 100 | 3.36 | 1.53 |
| 150 | 3.68 | 1.53 |
| 200 | 4.70 | 1.53 |

### A.1.9 TRAINING COMPLEXITY

While the previous section confirms MOSAIC's zero-overhead advantage during inference, we also provide a detailed analysis of the training computational cost relative to the number of sampled points ($P^{(k)}$).

As analyzed in the previous ablation study (Sec.A.1.5), increasing $P^{(k)}$ improves performance up to a saturation point. To balance these gains with efficiency, we measured the training time consumption per-step of the SCAL and MDL. As shown in Table 14, setting $P^{(k)} = 150$ (our default configuration) results in a manageable time overhead of 3.68 s/step, representing merely a **29.1% increase** in training cost compared to the baseline. In contrast, increasing to 200 points leads to a sharp spike in computation time (4.70 s/step) while yielding diminishing performance gains. This analysis justifies our choice of 150 points as the optimal trade-off between training efficiency and generation quality.

### A.2 MORE QUALITATIVE COMPARISONS

We provide extensive qualitative results in Fig. 15, Fig. 16 and Fig. 17 to further demonstrate the superiority of our approach in maintaining visual coherence across diverse multi-subject scenarios. Our method consistently outperforms state-of-the-art approaches in several critical aspects:

**Fine-grained Detail Preservation.** In row 1 of Fig. 16, while DreamO Mou et al. (2025a) and OmniGen2 Wu et al. (2025a) successfully preserve textual content on the bowl, only our method ac-

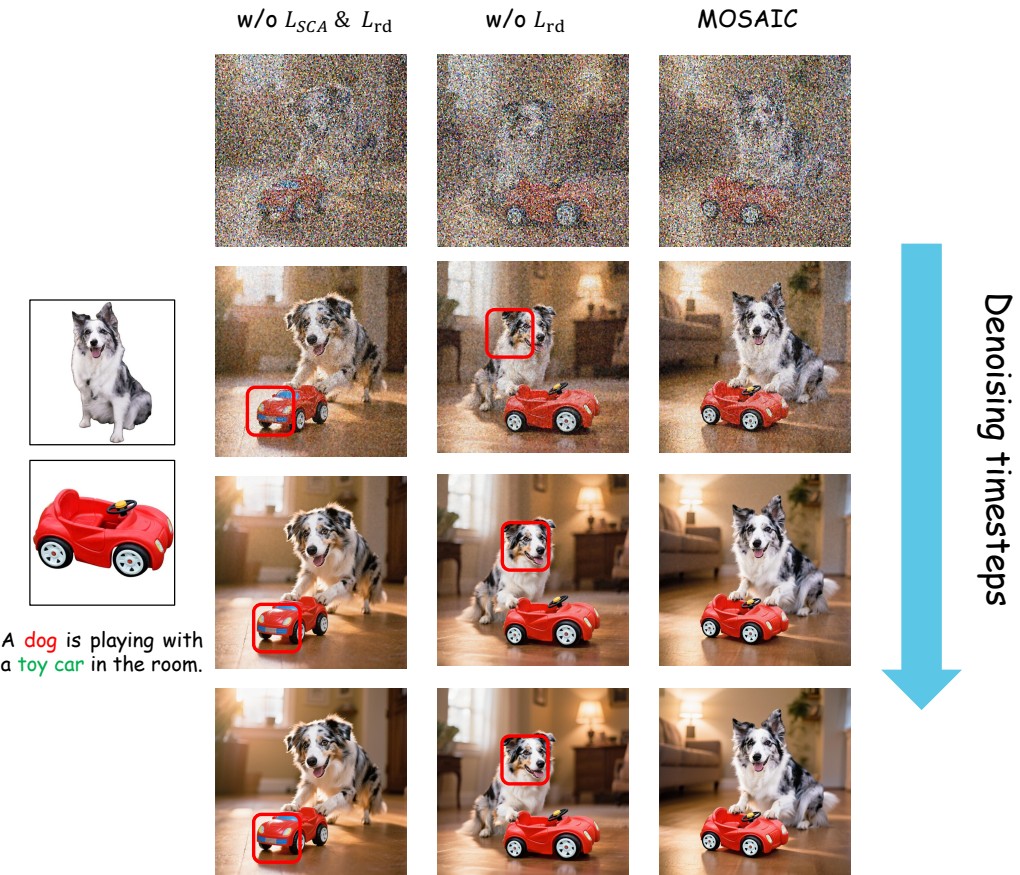

Figure 7: Step-by-step denoising visualization of mechanism effects. We compare the generation process across three settings: **(Left) Baseline**: Fails to preserve fine-grained identity details of the car. **(Middle) w/ $L_{SCA}$ only**: Successfully aligns identity features but suffers from **feature entanglement**, where the yellow color from the car bleeds onto the dog (highlighted by the red box). **(Right) MOSAIC**: With the addition of $L_{MD}$, the model achieves both precise identity preservation and spatial disentanglement, ensuring correct color attribution for both the dog and the car.

curately reconstructs both the intricate shape and surface texture of the vase, demonstrating superior fine-grained detail transfer capabilities.

**Subject Completeness and Consistency.** Row 3 of Fig. 16 highlights common failure modes in existing methods: MS-diffusion Wang et al. (2025) and XVerse Chen et al. (2025) suffer from object repetition or omission artifacts, while UNO and OmniGen2 exhibit significant facial consistency degradation. In contrast, our approach accurately renders all three reference objects while maintaining overall visual coherence.

**Cross-Subject Interference Mitigation.** Across all examples, our method demonstrates robust performance in preventing attribute bleeding between subjects, maintaining distinct identity characteristics for each reference entity without compromising generation quality.

A.3 SCALABILITY TO CHALLENGING MULTI-SUBJECT SCENARIOS.

Fig 17 presents challenging **4+ subjects** generation results that highlight the scalability advantages of our approach. In the superhero scene (row 1), existing methods struggle with identity confusion and incomplete subject rendering: MS-Diffusion Wang et al. (2025) loses Captain America's distinctive shield design, UNO Wu et al. (2025b) fails to maintain Iron Man's armor details, and XVerse Chen et al. (2025) exhibits significant pose distortions. Our method successfully preserves all four superheroes with accurate costumes, poses, and distinctive characteristics. Similarly, in

the multi-object scene (row 2), while competing methods suffer from object omission and attribute mixing (incorrect colors or textures bleeding between subjects), our approach maintains perfect subject completeness and identity consistency across all 5+ reference entities. This demonstrates our method's unique capability to scale beyond the typical 2-3 subject limitation of existing approaches, enabling complex real-world multi-subject generation scenarios.

## A.4 GENERATION DIVERSITY ANALYSIS

A common concern in personalization tasks is the trade-off between fidelity and diversity: stronger identity constraints often lead to mode collapse or rigid copying of reference poses. To demonstrate that MOSAIC achieves high fidelity without sacrificing diversity, we conduct both quantitative and qualitative evaluations.

**Quantitative Evaluation.** We performed a rigorous "intra-class diversity" test on the DreamBench multi-reference scenario. For each test case, we fixed the prompt and reference images, generated outcomes using 10 different random seeds, and calculated the average LPIPS distance. As shown in Table 15, MOSAIC achieves the highest diversity score (52.15), outperforming existing SOTA methods.

Table 15: Quantitative diversity comparison. We report the average LPIPS distance calculated across images generated with fixed prompts but different random seeds. A higher LPIPS score indicates greater perceptual diversity.

| Method | LPIPS ($\uparrow$) |
|---|---|
| MS-Diffusion | 45.85 |
| XVerse | 48.55 |
| DreamO | 50.10 |
| UNO | 51.20 |
| **MOSAIC (Ours)** | **52.15** |

**Qualitative Visualization.** As illustrated in Fig. 8, under fixed prompts and reference subjects, our model generates highly diverse outputs across varying seeds. For instance, the cat is generated with varying poses (sitting vs. lying down) and the desktop objects appear in different spatial layouts.

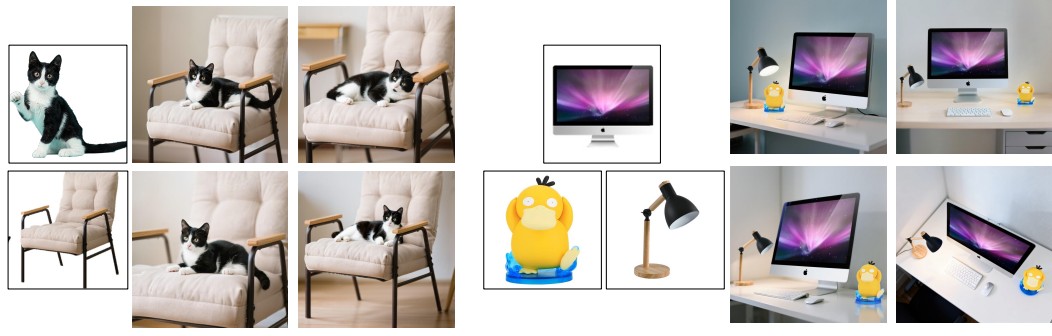

A cat lies on a chair in the living room.     A computer, a lamp and a toy on a desk

Figure 8: Qualitative evaluation of generation diversity. We fix the text prompts and reference images while varying random seeds. **(Left)** The model produces diverse **poses** for the cat and varying **viewpoints** for the chair. **(Right)** The **spatial layout** and lighting of the desktop scene vary significantly across samples. This confirms that MOSAIC preserves high subject fidelity without rigidly overfitting to the reference image geometry.

**Mechanism Analysis.** We attribute this superior balance to the nature of our **Point-wise Semantic Constraint**: Unlike global feature alignment strategies that constrain the entire latent space, our $L_{SCA}$ only anchors key semantic feature points. This leaves the diffusion model with significant freedom to generate non-critical areas such as backgrounds, lighting, and specific poses. Simultane-

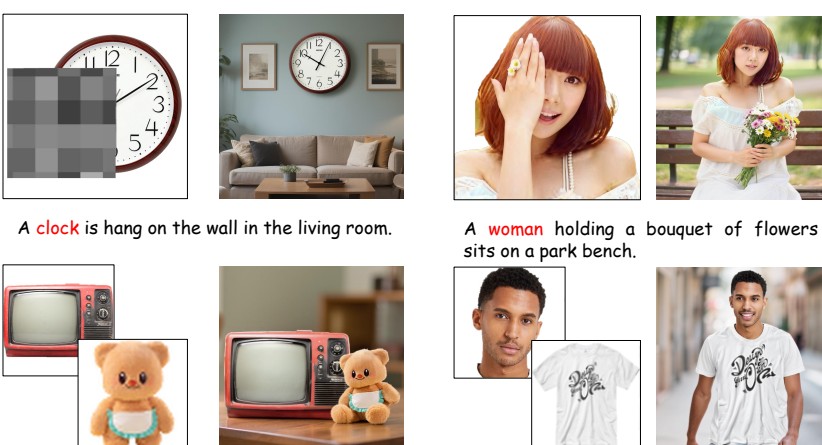

A clock is hang on the wall in the living room.

A woman holding a bouquet of flowers sits on a park bench.

A television is placed on the table with a bear toy next to it.

A smiling man in a t-shirt walk in the street.

Figure 9: Robustness evaluation under challenging input conditions. We test MOSAIC's performance when reference images are imperfect. **(Top Row) Partial Occlusion:** Even when reference subjects are significantly occluded (e.g., a masked clock or a face covered by a hand), the model successfully reconstructs complete, identity-consistent subjects. **(Bottom Row) Mixed Resolution:** In scenarios combining high-quality and low-quality inputs (e.g., a clear TV paired with a blurry teddy bear), the model generates all subjects with uniform high fidelity. This demonstrates the robustness of our semantic feature alignment mechanism against input degradation.

ously, $L_{MD}$ prevents feature entanglement between subjects, avoiding "mode collapse" where the model might otherwise revert to a confused, average representation.

## A.5 ROBUSTNESS TO CHALLENGING INPUTS: OCCLUSION AND MIXED RESOLUTION

Real-world applications often involve imperfect reference images, such as subjects being partially occluded or having varying resolutions. To evaluate MOSAIC's robustness under these challenging conditions, we conducted specific visualization experiments as shown in Fig. 9.

**Handling Partial Occlusion.** The top row of Fig. 9 demonstrates scenarios where reference subjects lose significant visual information (e.g., a clock partially masked by a gray patch, or a woman covering one eye). Despite these occlusions, MOSAIC successfully reconstructs the complete subjects with consistent identity. This capability suggests that our model does not mechanically copy pixels but rather anchors the subject's identity using **key semantic points from unoccluded regions**, effectively completing missing information.

**Handling Mixed Resolution.** The bottom row illustrates mixed-quality scenarios, such as pairing a high-resolution TV with a blurred, low-resolution teddy bear. Despite the disparity in input quality, the generated results exhibit uniform high fidelity for all subjects. This proves that our alignment mechanism can extract core semantic features even from degraded inputs, aligning them within a unified high-quality feature space.

## A.6 FAILURE CASE ANALYSIS AND LIMITATIONS

While MOSAIC demonstrates state-of-the-art performance across a wide range of benchmarks, we conduct a transparent analysis of failure cases to highlight current limitations. As illustrated in Fig. 10, our model faces challenges in scenarios involving extreme spatial complexity:

**Complex Physical Containment.** The first row of Fig. 10 presents a nested containment task: "A plushie toy inside a glass jar, which is placed inside a box." While MOSAIC successfully generates all three elements and correctly places the jar within the box, maintaining the **visual consistency**

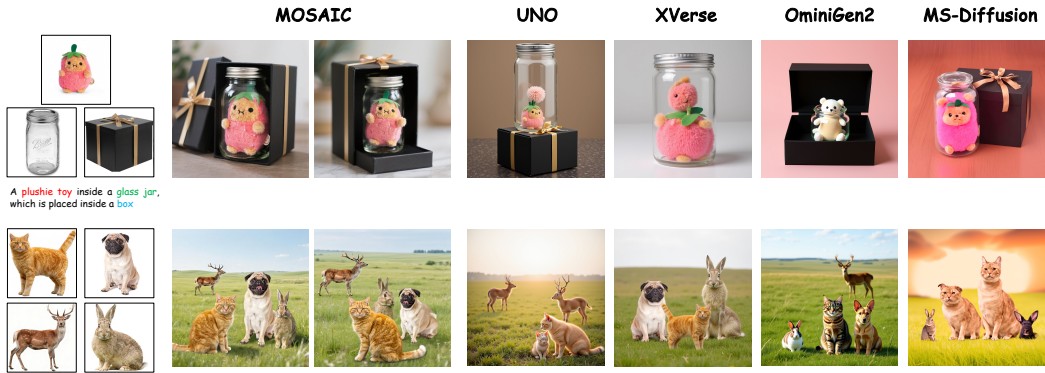

Figure 10: Failure case analysis in highly complex scenarios. We visualize two representative limitations: (Top Row) **Complex Physical Containment**, where the model captures the nested spatial relationship but suffers from **visual consistency degradation** of the outer container (the box); and (Bottom Row) **Fine-grained Spatial Layouts**, where strict adherence to complex positional constraints for multiple subjects is not perfectly maintained. Despite these limitations, MOSAIC still exhibits superior subject preservation and prompt adherence compared to baselines.

**of the outer container (the box)** remains challenging. Due to the complexity of the nested composition, the box occasionally exhibits structural distortions or a loss of texture details compared to the reference. However, compared to baseline methods like UNO and XVerse—which often fail to generate the container entirely or blend objects together—MOSAIC still achieves the best overall structural coherence.

**Fine-grained Spatial Constraints.** The second row demonstrates a complex spatial layout task requiring specific positioning of four animals (cat, dog, rabbit, deer). While MOSAIC generates all subjects with high fidelity, it occasionally fails to strictly adhere to rigorous geometric constraints (e.g., specific "left-right" alignment), resulting in minor semantic mismatches regarding position. Despite this, it significantly outperforms competitors that frequently suffer from severe subject omission or identity confusion in such high-density scenes.

## B DETAILED DATA CONSTRUCTION AND COMPARISON

### B.1 DATA CONSTRUCTION PIPELINE

Our data construction follows a systematic five-stage pipeline designed to generate high-quality multi-subject training pairs with validated semantic correspondences.

**Stage 1: Multi-Subject Prompt Generation.** We design structured image generation templates and leverage GPT-4o to produce diverse multi-object prompts. These prompts comprehensively span various combinations of humans, animals, objects, and their interactions, ensuring thorough coverage of complex multi-entity scenarios encountered in real-world applications.

**Stage 2: High-Quality Image Synthesis and Filtering.** The generated prompts are fed into state-of-the-art text-to-image models to synthesize corresponding images. To ensure data quality and fidelity, we implement a comprehensive multi-criteria filtering pipeline: (1) We employ a CLIP+MLP-based aesthetic assessment model Schuhmann (2022) to discard images below predefined aesthetic thresholds. (2) We utilize GroundingDINO Liu et al. (2023) to verify high-confidence detection of each entity mentioned in the prompt, filtering out instances where subject regions are too small or inadequately captured. (3) We compute CLIP-based prompt-image semantic similarity scores to remove images that fail to meet semantic alignment thresholds.

**Stage 3: Subject Detection and Segmentation.** We apply Lang-SAM Kirillov et al. (2023) for robust open-vocabulary detection and segmentation across all synthesized images. This enables

precise localization and extraction of all subjects regardless of semantic category, establishing a foundation for reliable correspondence matching in subsequent stages.

**Stage 4: Multi-View Augmentation.** To enhance dataset diversity and ensure comprehensive appearance coverage across varied viewpoints and poses, we incorporate FLUX Kontext Labs et al. (2025) for viewpoint correction while preserving semantic coherence.

**Stage 5: Semantic Correspondence Construction and Quality Assessment.** During image pair construction, we employ rigorous cross-validation to maximize correspondence reliability and semantic consistency. For each reference-target pair, we apply targeted masking to isolate reference subject regions in the target image, minimizing interference from other entities. We utilize both DIFT Tang et al. (2023) and GeoAware-SC Zhang et al. (2024a) to establish fine-grained token-level correspondences, aggregating tokens mapped to the same region into sampling pools.

**Stage 6: Dataset Quality Assessment.**

To rigorously ensure the quality and accuracy of our dataset, we implemented a comprehensive two-stage assessment mechanism:

**Automated Cross-Validation**. As detailed in Appendix B.1 (Line 786), we eschew reliance on a single model in favor of an "intersection strategy" to filter out noise. Specifically: We utilize two distinct feature extraction methods: DIFT Tang et al. (2023) and Geo-Aware Zhang et al. (2024a) features. A sample is retained only when the corresponding points extracted by both methods exhibit high spatial consistency (i.e., falling within the intersection region). This approach leverages the complementarity of different algorithms to automatically eliminate the vast majority of potential matching errors.

**Human-in-the-loop Verification**. To further quantify the dataset's accuracy, we conducted a rigorous manual inspection process: We performed stratified sampling on 10% of the full dataset, ensuring a balanced distribution across both single/multi-reference" scenarios and various object categories." Human annotators manually verified the semantic consistency of the matched points in these samples. The final statistics indicate that the error rate is strictly controlled within 3%.

Through this strict automated filtering and statistical human verification, we strongly guarantee the high quality and reliability of the SemAlign-MS dataset.

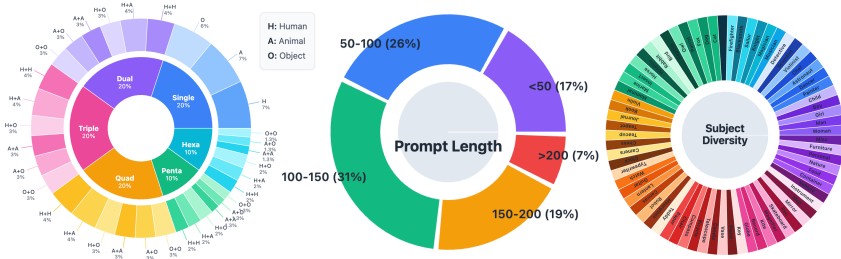

Figure 11: Detailed distribution and diversity statistics of the SemAlign-MS dataset. The visualizations illustrate the dataset's balanced construction across three key dimensions: (a) Reference Balance: The dataset maintains an even distribution of subject counts, ranging from single-subject scenarios to complex multi-subject compositions (up to six references). (b) Category Diversity: The data covers major semantic categories including Humans (H), Animals (A), and Objects (O), featuring rich cross-category interaction combinations (e.g., H+A, H+O). (c) Prompt & Subject Complexity: The distribution of prompt lengths demonstrates coverage from short labels (<50) to complex long-form descriptions (>200), alongside a wide variety of fine-grained subject classes

## B.2 DATASET STATISTICS AND VISUALIZATION OF FULL ANNOTATIONS

**Dataset Statistics.** Following these comprehensive assessment and filtering procedures, our pipeline yields a high-quality dataset comprising **1.2M reference-target image pairs** with validated semantic correspondences, providing precise supervision for multi-subject generation tasks. To comprehensively analyze the coverage and potential biases of our constructed dataset, we present

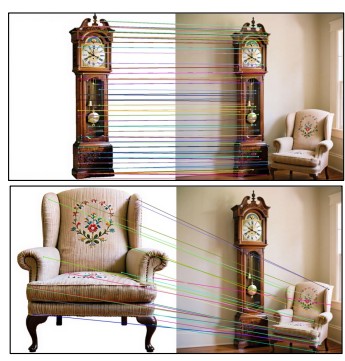
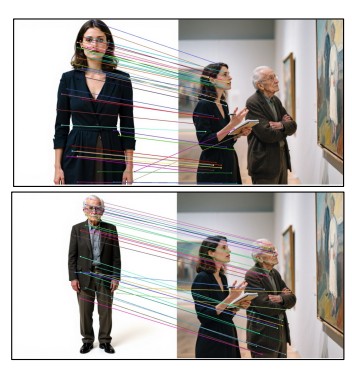
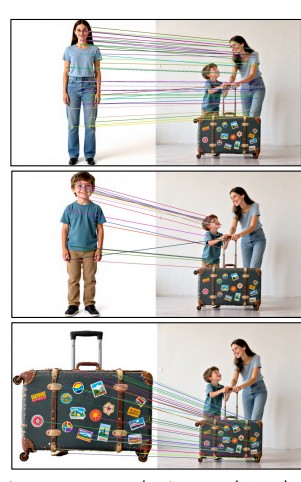

A tall grandfather clock stands beside a cozy upholstered armchair , creating a warm, inviting atmosphere of a comfortable home.

A woman and an elderly man stand at an art gallery, gazing at a large painting. She holds a notepad and gestures enthusiastically while discussing the artwork. He listens thoughtfully with arms crossed, offering quiet insights.

A young woman and a boy stand together, each gripping a handle of a large suitcase covered in colorful travel stickers. The boy looks up eagerly while the woman smiles with encouragement as they prepare to lift it.

Figure 12: Visualization of full annotation examples from SemAlign-MS. We showcase the dense semantic correspondences established by our pipeline across diverse scenarios: **(Left)** A dual-object scene (Grandfather Clock + Armchair); **(Middle)** A dual-human interaction (Woman + Elderly Man); and **(Right)** A complex triple-subject composition (Woman + Boy + Suitcase). The colored lines represent validated point-wise semantic matches between reference subjects and the generated target image. Crucially, notice how the correspondences are strictly isolated for each subject (e.g., separate mapping flows for the boy vs. the suitcase), ensuring precise multi-subject supervision without feature entanglement.

detailed distribution statistics in Figure 11. The statistics reveal that SemAlign-MS is constructed with a deliberate focus on balance and diversity across three critical dimensions:**(a) Reference Balance.** Unlike prior datasets that predominantly focus on single-subject generation, our dataset maintains a robust balance between single-subject and multi-subject scenarios. As shown in Figure 11(a), Single, Dual, and Triple-subject samples each constitute approximately 20% of the dataset, ensuring the model's capability to generalize across varying subject densities. We also include challenging high-density samples (Quad, Penta, and Hexa subjects) to further test the model's limits in complex composition.**(b) Category Diversity.** To prevent domain bias, we ensure a wide semantic coverage across three core categories: Human (H), Animal (A), and Object (O). Crucially, the dataset is not limited to isolated categories but features a rich array of cross-category interactions. The distribution in Figure 11(b) highlights substantial proportions of heterogeneous combinations (e.g., Human-Animal, Human-Object interactions), enabling the model to learn disentangled representations for diverse semantic concepts rather than overfitting to specific domains like faces.**(c) Prompt & Subject Complexity.** We also emphasize diversity in textual complexity and fine-grained subject classes. As illustrated in Figure 11(c), the prompt lengths follow a broad distribution, ranging from concise labels ($< 50$ tokens, 17%) to detailed, long-form descriptions ($100 - 150$ tokens, 31%; $> 200$ tokens, 7%). This variance ensures the model remains robust to different user prompting styles. Furthermore, the dataset encompasses a vast taxonomy of fine-grained subjects (e.g., specific professions, animal breeds, and object types), providing the necessary granularity for precise personalization.

**Visualization of Full Annotations.** To intuitively demonstrate the granularity and accuracy of our data annotations, we present detailed visualizations of the established semantic correspondences in Fig. 12. These examples illustrate how our automated pipeline generates dense, point-wise mappings for complex multi-subject scenarios. As shown, our method successfully establishes specific correspondences for each individual entity (e.g., distinguishing between the woman, the boy, and the suitcase) while maintaining strict semantic consistency. These high-quality, dense annotations serve as the foundational ground truth for our proposed point-wise alignment ($L_{SCA}$) and spatial disentanglement ($L_{MD}$) mechanisms.

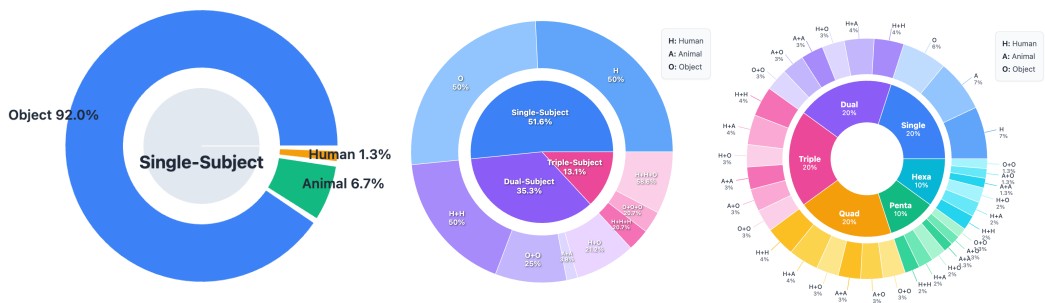

Figure 13: Comparative analysis of data distribution. We compare our SemAlign-MS with Subject-200K Tan et al. (2025) and X2I-subject-driven Xiao et al. (2024). **(Left)** Subject-200K is strictly limited to single-subject samples with a strong object-centric bias. **(Middle)** X2I-subject-driven supports up to three subjects but shows skewed distributions in category combinations (e.g., dominating H+H+O patterns). **(Right)** Our **SemAlign-MS** demonstrates superior balance, featuring a uniform distribution across subject counts (1 to 6) and diverse, unbiased semantic interactions.

### B.3 COMPARISON WITH EXISTING DATASETS.

To demonstrate the uniqueness and superiority of SemAlign-MS, we conduct a comparative distribution analysis against two representative datasets: Subject-200K Tan et al. (2025) (a widely used single-subject dataset) and X2I-subject-driven Xiao et al. (2024) (a multi-subject dataset). As visualized in Fig. 13:

- **Subject-200K (Left)** is predominantly composed of single-subject samples (100%), with a heavy bias towards the *Object* category (92.0%), lacking human or animal interactions.
- **X2I-subject-driven (Middle)** introduces multi-subject scenarios but is limited to a maximum of three subjects. Moreover, it exhibits significant bias in category combinations; for instance, in triple-subject scenarios, the specific "Human+Human+Object" combination dominates (58.6%), limiting the diversity of semantic interactions.
- **SemAlign-MS (Right)** achieves a significantly more balanced distribution. It covers a wide range of subject counts from 1 to 6 (including challenging high-density scenes like Hexa-subject) with an even ratio (~20% for 1-3 subjects). Furthermore, it features a rich and uniform distribution of cross-category interactions (e.g., Human-Animal, Animal-Object) without over-relying on specific patterns.

This comparison confirms that SemAlign-MS offers superior diversity and balance, providing a more robust foundation for training generalizable multi-subject generation models.

## C DETAILED STATISTICS OF EVALUATION BENCHMARKS

To ensure a fair and unbiased assessment of model performance, we utilize two carefully curated benchmarks: **DreamBench** and **XVerseBench**. As illustrated in Fig. 14, both benchmarks are constructed with strict controls on sample size, category distribution, and prompt diversity to eliminate potential data bias.

**DreamBench Statistics (Left).** DreamBench is strictly balance the ratio between single-subject and multi-subject scenarios. As shown in the pie chart, the dataset is split evenly:

- **Single-Subject (50%):** Comprises *Object* (35.0%) and *Animal* (15.0%) categories, covering fundamental subject generation capabilities.
- **Multi-Subject (50%):** Evenly distributed across three interaction types: *Animal+Animal*, *Animal+Object*, and *Object+Object* (16.7% each).

This 50/50 split ensures that metrics are not skewed by simple single-subject cases, providing a rigorous test for multi-subject consistency.

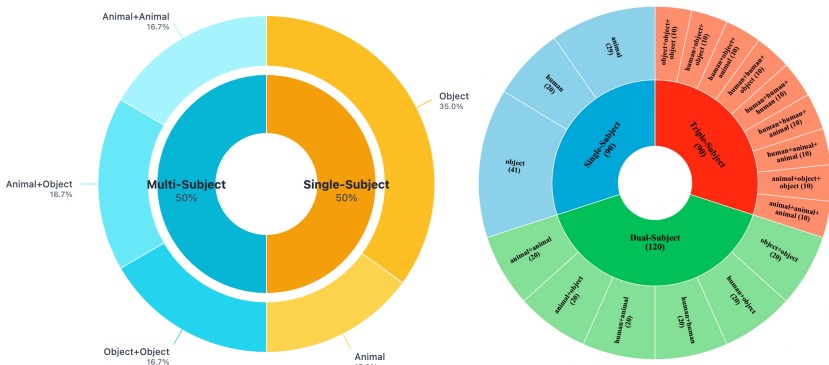

Figure 14: Statistical distribution of evaluation benchmarks. **(Left) DreamBench:** The benchmark maintain a strict 50%-50% balance between Single-Subject and Multi-Subject scenarios to ensure unbiased evaluation of basic and compositional capabilities. **(Right) XVerseBench:** The benchmark comprises 300 test cases with a difficulty gradient (Single/Dual/Triple subjects). It features a highly uniform distribution across diverse semantic categories (Human, Animal, Object) and their combinations, preventing evaluation bias towards specific domains.

**XVerseBench Statistics (Right).** Designed to evaluate complex multi-subject interactions, XVerseBench consists of 300 distinct test cases structured with a difficulty gradient:

- **Sample Size Distribution:** The benchmark includes 90 Single-Subject, 120 Dual-Subject, and 90 Triple-Subject samples, testing the model's scalability from 1 to 3 subjects.
- **Category Combination Balance:** To avoid bias towards specific domains (e.g., human faces), we ensure a uniform distribution of category combinations. For instance, the Dual-Subject set (120 cases) is equally divided into 6 combinations (e.g., Human+Human, Human+Object, Animal+Object, etc.), with 20 cases each. Similarly, the Triple-Subject set covers diverse permutations (e.g., H+H+A, H+A+A) to comprehensively assess interaction handling.

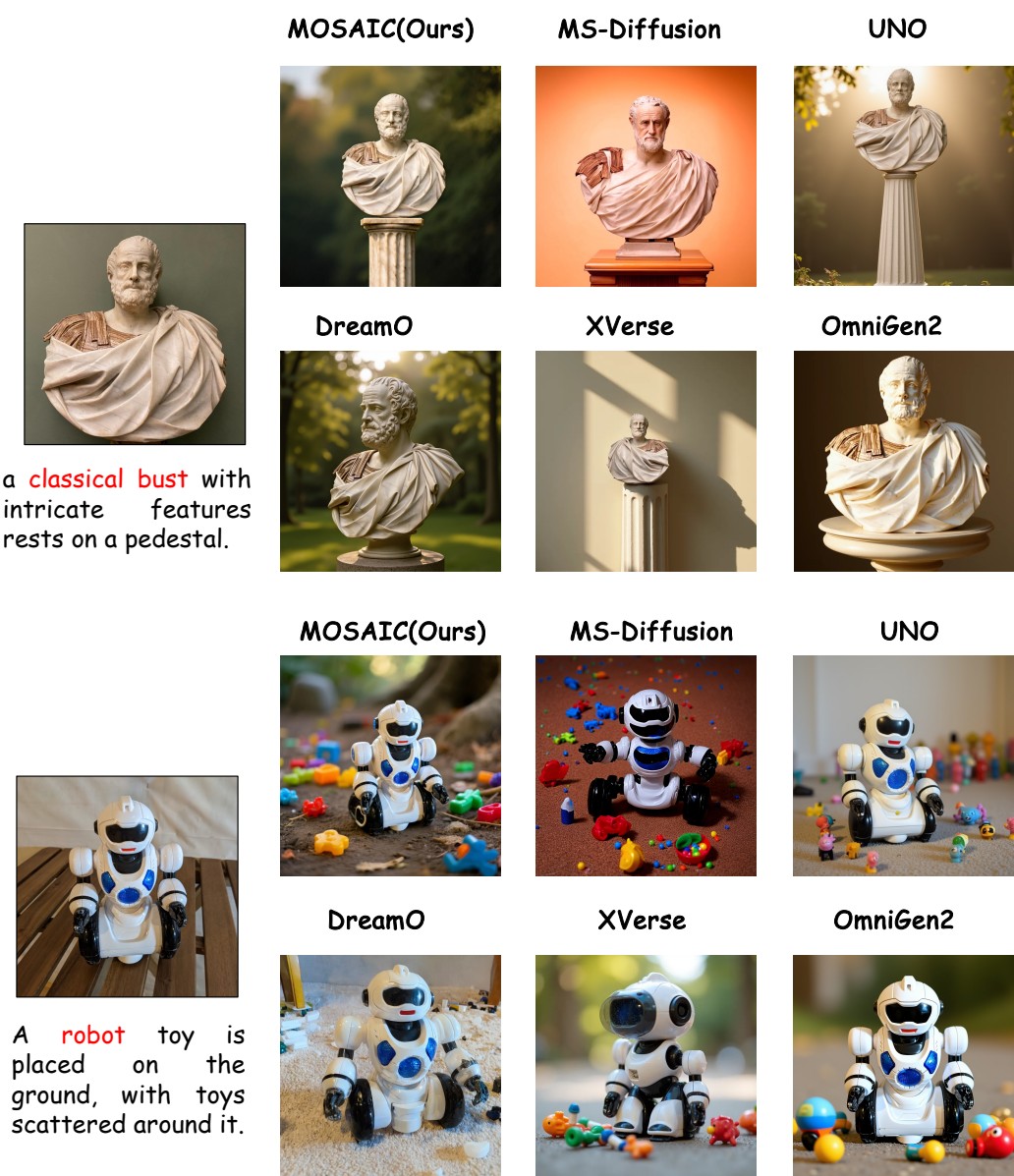

Figure 15: Qualitative comparison on single subject driven generation.

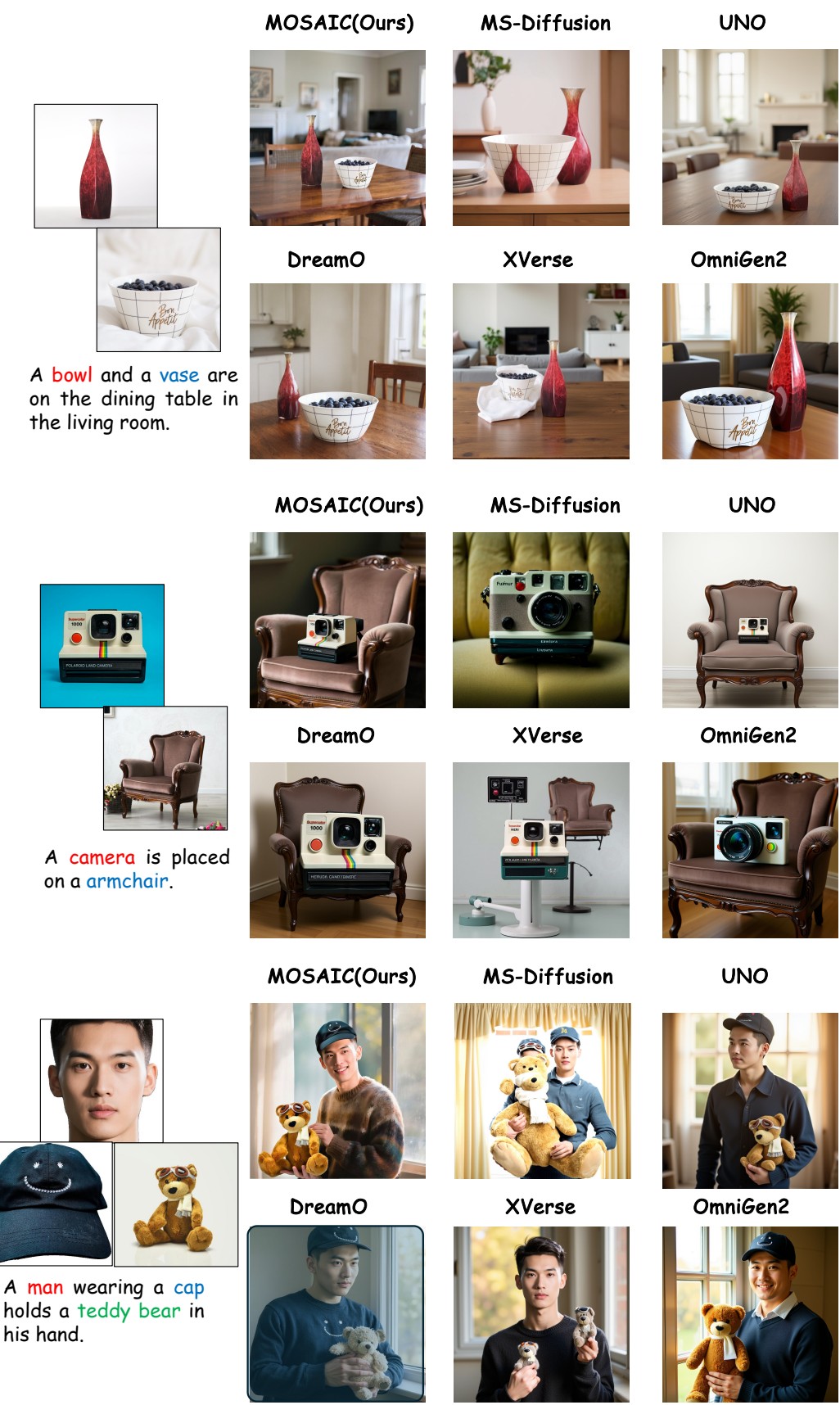

Figure 16: Qualitative comparison on multi-subject driven generation.

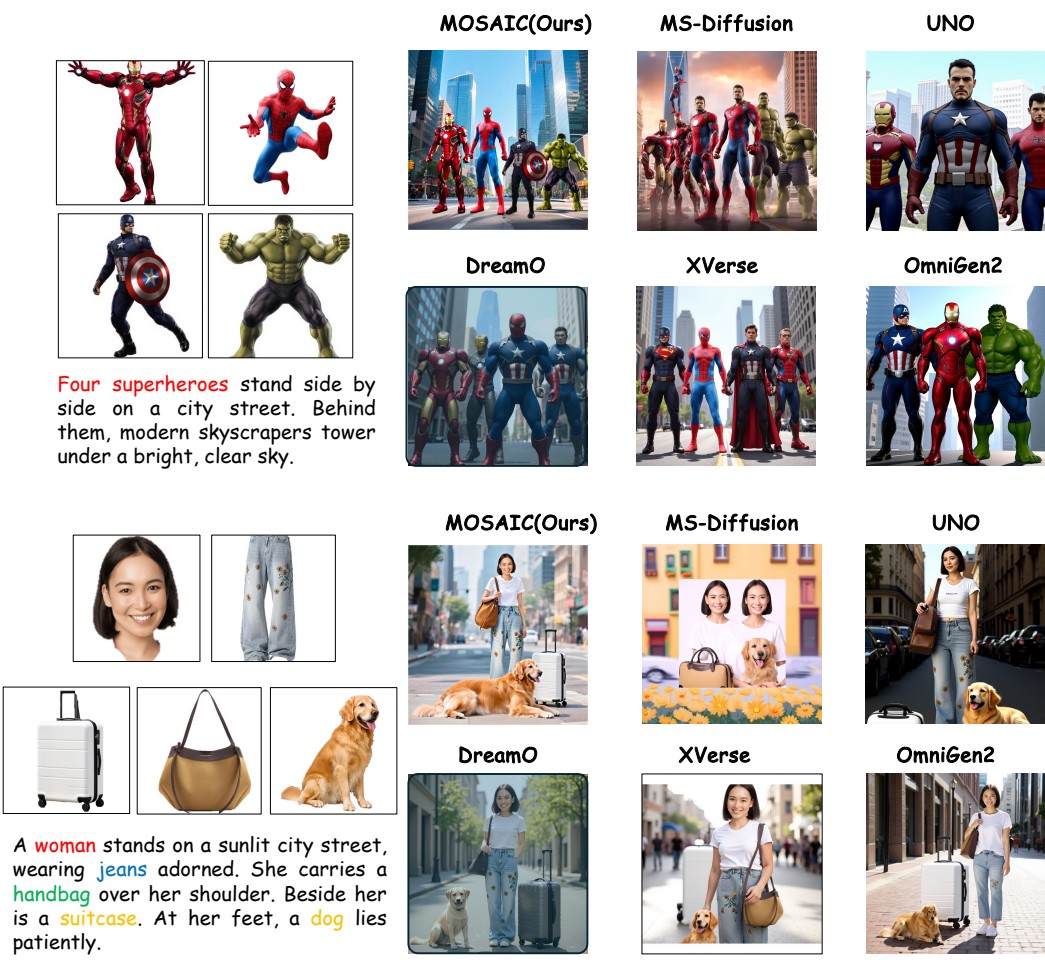

Figure 17: Qualitative comparison on 4+ subject driven generation.

