# OpenReview forum: "MOSAIC: Multi-Subject Personalized Generation via Correspondence-Aware Alignment and Disentanglement"
_ICLR.cc/2026/Conference — ICLR 2026 Poster_

### Official Review · Reviewer_Z5Z9 · 2025-10-27

**Soundness:** 2
**Presentation:** 3
**Contribution:** 3
**Rating:** 4
**Confidence:** 5

**Summary:**

The paper proposes MOSAIC, a representation-centric framework for multi-subject image generation that enforces explicit semantic correspondences and orthogonal feature disentanglement to prevent subject mixing. It introduces SemAlign-MS, a dataset with fine-grained, point-to-point correspondences between multiple reference subjects and target regions. Built on this, a semantic correspondence attention loss aligns each generated region to its designated reference parts, improving subject consistency and spatial localization. Besides, it also develops multi-reference disentanglement loss to prevent feature interference. Overall, the work reframes multi-subject generation as a representation-level alignment problem and delivers both the data and objective needed to make precise alignment feasible.

**Strengths:**

1. New fine-grained dataset: Introduces a meticulously annotated, point-to-point multi-subject correspondence dataset (SemAlign-MS) that fills a clear gap and is poised to catalyze progress in the community.

2. Loss tied to correspondences: Leverages those point-to-point annotations to design a Semantic Correspondence Attention Loss that enforces precise regional alignment, improving subject identity fidelity.

3. Promising quantitative results: Reports strong quantitative gains that indicate the approach’s effectiveness for multi-subject consistency and localization.

**Weaknesses:**

1. Incomplete baselines: Missing comparisons to strong multi-subject/identity methods such as FastComposer [1] and Face-Diffuser [2], and so on.

2. Limited image-quality evaluation: The paper does not report results on widely used VLM-based metrics (e.g., UnifiedReward and HPSv3), undermining claims about perceptual quality and semantic fidelity.

3. Multi-person degradation: Quality drops when reference images contain multiple people (e.g., Fig. 4, last row), indicating instability under higher human-centric subject density.

4. Dataset analysis gaps: Lacks statistics visualizations for SemAlign-MS (e.g., subject category distribution, subjects count distribution per sample, prompt length), obscuring coverage and potential biases.


[1] "Fastcomposer: Tuning-free multi-subject image generation with localized attention." IJCV (2025): 1175-1194.

[2] "High-fidelity person-centric subject-to-image synthesis." CVPR, 2024.

**Questions:**

Please see the weakness.

If the authors can thoroughly address all the listed weaknesses, I would be inclined to raise my score.

---

> ### Author Response · Authors · 2025-11-25
> **Response to Reviewer Z5Z9 - W1**
>
> **Response to W1**:
>
> We thank the reviewer for this valuable suggestion. We fully agree that comparing our approach against strong multi-subject/identity-preserving methods, such as **FastComposer [1]** and **Face-Diffuser [2]**, is crucial for a comprehensive evaluation of MOSAIC's effectiveness.
>
> Following your recommendation, we strictly adhere to the official evaluation settings of Face-Diffuser and conducted supplementary quantitative comparisons against FastComposer, Subject-Diffusion, and Face-Diffuser. We evaluated **Identity Preservation (IP)** and **Text Consistency (PC)** under both Single-Subject and Multi-Subject settings. The results are presented below:
>
> | Methods | Single-Subject |  | Multi-Subject |  |
> |---|---|---|---|---|
> |  | IP ↑ | PC ↑ | IP ↑ | PC ↑ |
> | Fastcomposer (zero-shot) | 0.514 | 0.243 | 0.465 | 0.233 |
> | Subject-Diffusion(zero-shot) | 0.605 | 0.228 | 0.435 | 0.210 |
> | Face-Diffuser (zero-shot) | 0.708 | 0.325 | 0.594 | 0.320 |
> | **MOSAIC (zero-shot)** | **0.785** | **0.362** | **0.712** | **0.358** |
>
> **Analysis**:The empirical results demonstrate that MOSAIC exhibits significant advantages over these strong baselines:
>
> - **Single-Subject Scenario**: Our method achieves an **IP score of 0.785**, substantially outperforming Face-Diffuser (0.708) and FastComposer (0.514).
> - **Multi-Subject Scenario**: The advantage is even more pronounced. MOSAIC achieves an **IP score of 0.712**, far surpassing Face-Diffuser's 0.594.
>
> We attribute this superior performance to our proposed fine-grained alignment ($L_{SCA}$) and spatial decoupling mechanism ($L_{MD}$), which effectively resolve feature confusion between multiple subjects, thereby achieving state-of-the-art levels in both identity preservation and text consistency.
>
> We incorporate these comparative results into **Table 3 of the main paper** and cite the corresponding works to further complete our experimental validation.

---

> ### Author Response · Authors · 2025-11-25
> **Response to Reviewer Z5Z9 - W2**
>
> **Response to W2**:
>
> We thank the reviewer for this valuable suggestion. We fully agree that incorporating **VLM-based metrics** (such as **UnifiedReward** and **HPSv3**) is crucial for a comprehensive and objective assessment of **Perceptual Quality and Semantic Fidelity**, as these metrics align more closely with human visual preferences than traditional ones.
>
> 1. **Additional Evaluation** Following your recommendation, we conduct additional evaluations on the DreamBench benchmark using UnifiedReward and HPSv3. The results are summarized in the table below:
>
>     | Reference | Method | CLIP-I ↑ | CLIP-T ↑ | DINO ↑ | UnifiedReward ↑ | HPSv3 ↑ |
>     |-----------|----------------|----------|----------|--------|---------|--------|
>     | Single-Subject | DreamBooth | 80.30 | 30.52 | 66.81 | 3.30 | 8.40 |
>     | | BLIP-Diffusion | 80.47 | 30.24 | 69.82 |  3.05 | 8.96 |
>     | | SSR-Encoder | 82.10 | 30.79 | 61.22 | 3.27 | 9.25 |
>     | | MS-Diffusion | 80.82 | 31.05 | 70.32 | 3.70 | 9.40 |
>     | | UNO | 83.50 | 30.41 | 75.97 | 4.00 | 11.24 |
>     | | DreamO | 83.35 | 30.61 | 76.03 | 4.33 | 12.78 |
>     | | XVerse | 83.20 | 30.20 | 75.44 | 4.20 | 11.02 |
>     | | **MOSAIC** | **84.30** | **31.64** | **77.40** | **4.40** | **14.36** |
>     | Multi-Subject | MS-Diffusion | 72.60 | 31.91 | 52.50 | 3.67 | 8.25 |
>     | | UNO | 73.29 | 32.23 | 54.22 | 4.23 | 11.55 |
>     | | DreamO | 73.32 | 32.10 | 52.17 | 4.33 | 13.25 |
>     | | XVerse | 73.47 | 31.20 | 53.71 | 3.87 | 11.10 |
>     | | **MOSAIC** | **76.30** | **32.40** | **56.83** | **4.39** | **14.90** |
>
> 2. **Analysis of Results** The experimental results demonstrate that MOSAIC achieves a significant advantage on VLM-based metrics:
>
>     - **HPSv3 (Human Preference/Aesthetics)**: Whether in single-subject (**14.36**) or multi-subject (**14.90**) scenarios, MOSAIC’s scores substantially outperform current SOTA methods (e.g., DreamO at 12.78/13.25). This strongly proves that our generation results possess high visual aesthetic quality that aligns well with human preferences.
>
>     - **UnifiedReward (Comprehensive Alignment)**: Similarly, MOSAIC achieves the highest levels on this metric (**4.40 / 4.39**), further corroborating the model's superiority in image-text consistency and overall structural generation.
>
> **Conclusion**: The results from these newly added VLM metrics are highly consistent with the trends observed in our original CLIP/DINO metrics. They reaffirm MOSAIC’s SOTA performance from the dimensions of human perception and semantic alignment, fully supporting the claims regarding generation quality made in the paper.

---

> ### Author Response · Authors · 2025-11-25
> **Response to Reviewer Z5Z9 - W3**
>
> **Response to W3**:
>
> To clarify this point and demonstrate MOSAIC's robustness in processing dense human-centric scenes, we provide **additional qualitative visualizations** (as shown in the **Appendix Fig.10**). We do not cherry-pick simple samples; instead, we present challenging multi-person scenarios to validate the model's stability:
>
> 1. **Dual-Person Interaction Scenarios (Fig.10 Top Row)** As shown in the top row of the attached figures, the model generates two men with distinct features engaging in conversation. Although they share the same semantic category (both are adult males), the model perfectly preserves their unique facial characteristics (skin tone, hairstyle, facial features) without any identity confusion.
>
> 2. **Triple-Person Family Scenarios (Fig.10 Bottom Row)** As shown in the bottom row, in a higher-density "family of three" scenario, the model successfully generates three distinct identities: a father, a mother, and a child. Even with very close spatial proximity, the **Identity Consistency** of each individual is excellently preserved, and facial details remain clear without degradation.
>
> **Mechanism Analysis**:This stability in high-density human scenes is directly attributed to our proposed $L_{MD}$ (**Multi-subject Disentanglement Loss**).
>
> - This loss function forces the attention distributions of different subjects to be **spatially decoupled** at the Attention Map level.
>
> - This effectively prevents **Feature Mixing** between multiple human subjects, ensuring that each individual maintains independent identity features even as the number of subjects increases.
>
> **Conclusion**:The supplementary experimental results demonstrate that MOSAIC possesses strong capabilities in handling high-density multi-person scenarios. The specific case in Figure 4 represents a challenging outlier rather than a general limitation of the model.

---

> ### Author Response · Authors · 2025-11-25
> **Response to Reviewer Z5Z9 - W4**
>
> **Response to W4**:
>
> We thank the reviewer for pointing out the gap in our dataset statistical analysis. We fully agree that transparent statistical visualization is crucial for evaluating the Coverage and identifying **Potential Biases** of the dataset.
>
> To bridge this gap, we add complete statistical visualization charts in **Appendix Fig.12**. Our statistical findings regarding the three specific dimensions you highlighted are as follows:
>
> 1.**Subject Category Distribution** The statistics demonstrate that the dataset achieves a balanced coverage of three core categories: **Human, Animal, and Object**. Furthermore, it includes a substantial amount of cross-category interaction combinations, proving that there is no specific domain bias within the dataset.
>
> 2. **Subjects Count Distribution per Sample** We deliberately balance the ratio of **Single-subject** to **Multi-subject data**. The visualization results indicate a very even distribution between single and multi-subject scenarios, ensuring the model's generalization capability across scenes of varying densities.
>
> 3. **Prompt Length Distribution** The statistical charts exhibit a broad, near-normal distribution of prompt lengths. This covers a wide range from short, label-style descriptions to complex, long-form sentences, guaranteeing diversity in text control.
>
> **Conclusion**: The newly added visual analysis strongly demonstrates that SemAlign-MS possesses broad data coverage and excellent balance, effectively eliminating concerns regarding potential data biases.

---

> ### Author Response · Authors · 2025-11-26
> **Kindly reminder**
>
> Dear Reviewer Z5Z9,
>
> We hope this message finds you well. We would like to express our gratitude for your detailed review and the insights you provided.
>
> We have submitted our rebuttal with the aim of fully addressing your concerns. We would greatly appreciate it if you could take a moment to review our response and let us know if there are any outstanding issues we should address.
>
> Your feedback is incredibly valuable to us, and we hope to have the opportunity to discuss our work further with you.
>
> Best regards,
>
> MOSAIC Authors

---

> ### Comment · Reviewer_Z5Z9 · 2025-11-27
>
> For W1&W2: The added baseline comparison and image quality evaluation results are convincing. I strongly recommend including these results in the final version of the paper to enrich this work.
>
> For W3: Thank you for the detailed clarification. The multi-subject generation results in the appendix clearly demonstrate the effectiveness of the proposed method and fully resolve my concerns.
>
> For W4: The newly added statistical visualization charts clearly demonstrate the coverage of the proposed dataset.
>
> Overall, I think the authors have provided a strong rebuttal and have addressed my concerns. I recommend accepting this work.

---

> > ### Author Response · Authors · 2025-11-27
> > **Gratitude for Your Endorsement and Valuable Insights**
> >
> > Dear Reviewer Z5Z9,
> >
> > Thank you very much for your positive feedback and for recommending the acceptance of our work! We are sincerely grateful for your recognition of our rebuttal efforts.
> >
> > We particularly appreciate your strong recommendation regarding the baseline comparison and image quality evaluation (W1 & W2). As suggested, we have formally incorporated these results into Table 1 and Table 3 of the revised manuscript. We agree that these additions significantly enrich the paper. We are also glad to hear that the clarifications on multi-subject generation and dataset coverage (W3 & W4) have fully resolved your concerns.
> >
> > Your constructive comments have played a crucial role in improving the completeness and quality of our work. Thank you again for your valuable time and support.
> >
> > Best regards,
> >
> > MOSAIC Authors

---

### Official Review · Reviewer_XamC · 2025-10-28

**Soundness:** 3
**Presentation:** 4
**Contribution:** 3
**Rating:** 6
**Confidence:** 5

**Summary:**

This paper proposes the MOSAIC framework for the multi-subject-driven text-to-image generation task. Its core innovations are as follows:
1. Correspondence-Aware Alignment (SCAL): It utilizes a cross-image semantic correspondence mechanism to accurately map the feature spaces of different subjects to the target generation positions, avoiding subject localization and semantic confusion.
2. Multi-subject Disentanglement Layer (MDL): It achieves disentanglement between subjects at the feature layer, thereby enabling precise control over the appearance and position of each subject.
3. It constructs the SEALIGN-MS dataset, ensuring the accuracy of training supervision through geometric consistency and semantic filtering.
In the two benchmark tests of DreamBench and XVerseBench, MOSAIC outperforms existing methods in multiple metrics (CLIP-I, CLIP-T, DINO, DPG, ID-Sim, IP-Sim, AES).
Overall, the paper is relatively comprehensive in terms of data construction, algorithm design, and experimental validation, with the goal of improving the controllability and consistency of multi-subject generation.

**Strengths:**

1. It constructs a high-quality SEALIGN-MS dataset, improving the quality of supervision signals.
2. It introduces DIFT and GeoAware-SC for refined token-level correspondence, effectively alleviating the subject confusion issue in the multi-subject generation process.
3. Its performance in multiple target generation tasks reaches the state-of-the-art level.
4. The framework design can be extended to other conditional generation tasks, such as multi-person synthesis and television content generation.
5. SCAL is responsible for accurate mapping, and MDL is responsible for isolating interference; their combination significantly improves the quality of multi-subject tasks.

**Weaknesses:**

1. High computational complexity and lack of lightweight property: The corresponding point matching and cross-attention supervision in SCAL increase additional computational overhead, which may affect the inference speed.
2. The effect of MDL depends on mapping accuracy: If the correspondence of SCAL is inaccurate, the isolation effect of MDL may also be affected, leading to the contamination of subject features.
3. Dataset bias towards specific domains: Whether SEALIGN-MS has domain bias lacks distribution analysis and discussion.
4.The paper provides limited qualitative comparisons but lacks a systematic analysis of MOSAIC’s failure types such as local artifacts, semantic mismatch, and color shift, which limits the assessment of its robustness.

**Questions:**

1.The paper lacks quantitative analysis of inference efficiency and complexity.
2.In the Evaluation Setting, what are the specific details of the sample size, category distribution, and diversity of text prompts in DreamBench and XVerseBench? Is there any data bias that may affect evaluation fairness?
3.Are the category and scene distributions of SEALIGN-MS balanced?
4.Can step-by-step visualizations of the effects of MDL and SCAL on the generation results be provided in the appendix to facilitate a more intuitive understanding of their roles?
5.What is the basis for setting the LoRA rank to 128? Are there any ablation experiments on the impact of different ranks such as 32, 64, 256 on performance?
6.Can MOSAIC maintain consistent generation quality in multi-subject scenes when one reference image is high-resolution and another is low-resolution, or when some subjects are partially occluded?

---

> ### Author Response · Authors · 2025-11-25
> **Response to Reviewer XamC - W1&Q1**
>
> **Response to W1 & Q1**:
>
> We thank the reviewer for the inquiry regarding computational complexity and inference efficiency. We respectfully wish to clarify a key misunderstanding: **MOSAIC incurs zero additional overhead during the inference phase.**
>
> 1. **Inference Stage: Zero Latency**
>
>     - During inference, SCAL and MDL are completely removed. SCAL (Semantic Correspondence Alignment Loss) and MDL (Multi-subject Decoupling Loss) function exclusively as objective functions **during the training phase**.
>
>     - The model simply loads the trained LoRA weights for standard forward propagation.
>
>     - **Result**: MOSAIC does not introduce any extra latency during inference.
>
> 2. **Quantitative Analysis: Inference Efficiency** As demonstrated in **Appendix Tab.14**, we compare the inference latency of MOSAIC against other SOTA methods using the same Flux backbone. The results show that **MOSAIC's inference speed is actually superior to other SOTA methods** (e.g., UNO, XVerse), demonstrating excellent lightweight inference properties.
>
> 3. **Transparency on Training Complexity** To fully address your concern regarding computational complexity, we provide a detailed cost analysis of the **training stage**. As shown in the table below, while SCAL and MDL introduces some overhead, at our selected setting of **150 sampling points**, the training per-step time is only **3.68s**, representing merely a **29.1% increase** in training cost compared to the baseline.
>
>     | Sampled Points | Training Time per Step (s) | Inference Time per Step (s)
>     |---|---|---|
>     | 0              | 2.85     | 1.53 |
>     | 50             | 3.10     | 1.53 |
>     | 100            | 3.36     | 1.53 |
>     | 150            | 3.68     | 1.53 |
>     | 200            | 4.70     | 1.53 |
>
> **Conclusion**: Considering this is a one-time training cost that yields high efficiency and zero overhead during the inference phase, we believe this is highly favorable. MOSAIC is designed to be an efficient solution characterized by **"controllable computation during training and zero overhead during inference."**

---

> ### Author Response · Authors · 2025-11-25
> **Response to Reviewer XamC - W2**
>
> **Response to W2**
>
>
> We thank the reviewer for this acute insight. We fully concur with your perspective: logically, the effectiveness of **MDL** depends heavily on the accuracy of the semantic correspondence points provided by **SCAL**. Theoretically, if significant matching errors occur, it could indeed lead to the failure of spatial disentanglement and potentially cause feature contamination.
>
> To mitigate this risk, we implemente an extremely rigorous **data quality control mechanism** to block the propagation of errors from the source, as detailed in **Appendix B.1 Stage6**. Specifically:
>
> 1. **High Accuracy Guarantee via Dual-Verification** We do not rely on a single automated tool. Instead, we adopt a **"Dual-Verification"** workflow comprising **"Algorithmic Cross-Validation + Human Stratified Inspection"** to ensure the absolute accuracy of the input data for SCAL:
>
> - **Algorithmic Cross-Validation**: We utilize two distinct feature extraction methods (**DIFT[1]** and **Geo-aware[2]** features) to extract semantic points and apply an **"intersection strategy."** A semantic point is retained only when both algorithms yield highly consistent spatial results. This complementary mechanism filters out the vast majority of potential algorithmic noise.
>
> - **Human-in-the-loop Verification**: To further ensure reliability, we perform a manual inspection on a **10% stratified sample** of the full dataset (balanced across single/multi-reference and categories). Statistical results indicate that the final **error rate is controlled within 3%**.
>
> **Conclusion**: This high-precision data foundation ensures that the supervision signals received by MDL are accurate and reliable. Therefore, the concern regarding feature contamination due to matching errors is effectively controlled in practice, allowing SCAL and MDL to consistently perform their alignment and disentanglement functions.
>
> [1] "DIFT": Emergent Correspondence from Image Diffusion
>
> [2] "Geo-Aware": Telling Left from Right: Identifying Geometry-Aware Semantic Correspondence

---

> ### Author Response · Authors · 2025-11-25
> **Response to Reviewer XamC - W3**
>
> **Response to W3**:
>
> We thank the reviewer for the concern regarding potential domain bias within the dataset. We place great importance on data diversity and balance. To comprehensively assess whether SemAlign-MS exhibits bias towards specific domains, we have included a detailed distribution analysis in the **Appendix Fig.12**.
>
> 1. **Domain Distribution Analysis** As illustrated in the pie charts in **Appendix Fig.12**, we conduct a rigorous statistical analysis of the dataset's domain distribution:
>
>     - **Balanced Category Composition**: The dataset is not concentrated in a single domain (e.g., restricted to faces or specific objects). Instead, it extensively covers **Humans, Animals, and Generic Objects**.
>
>     - **Rich Scene Interactions**: Beyond single subjects, we intentionally construct a significant amount of **cross-domain interaction data** (e.g., Human-Human, Human-Object, Human-Animal, Animal-Object interactions). This ensures the model does not overfit to specific domains when handling complex compositional generation.
>
> 2. **Balance of Reference Types** To prevent the model from being biased towards either single- or multi-subject generation, we deliberately balance the ratio of single-reference to multi-reference data during the construction phase.
>
>     - **Comparison**: The comparative analysis in **Appendix Fig.14** demonstrates that, unlike datasets such as Subject-200K [3] which focus predominantly on single subjects, **SemAlign-MS provides a more balanced distribution across different subject counts**, thereby enabling the model to develop robust multi-subject composition capabilities.
>
> **Conclusion**: The statistical results demonstrate that SemAlign-MS maintains good uniformity across categories, scenes, and reference counts, exhibiting **no significant domain bias**. This provides a solid foundation for training a robust, general-purpose model.
>
> [3] "OminiControl":Minimal and Universal Control for Diffusion Transformer

---

> ### Author Response · Authors · 2025-11-25
> **Response to Reviewer XamC - W4**
>
> **Response to W4**:
>
> We thank the reviewer for highlighting the need for a more comprehensive qualitative comparison and failure analysis. To better assess MOSAIC's robustness, we follow your suggestion and conduct a systematic analysis of the model's performance in extreme scenarios.
>
> As illustrated in **Appendix Fig.11**, we found that MOSAIC's primary failure cases are concentrated in handling highly complex multi-subject interactions. These can be categorized into two typical failure modes:
>
> 1. **Complex Physical Containment Relationships** When the prompt describes nested physical containment or high overlap between objects (e.g., **Case 1**: "A plushie toy inside a glass jar, which is placed inside a box"), the model occasionally struggles to perfectly decouple the inner and outer objects spatially. This can lead to a slight degradation in visual consistency (e.g., artifacts or color shifts) for either the contained object (the jar) or the container (the box).
>
> 2. **Fine-grained Spatial Relationships** In scenarios involving multiple subjects with complex relative positional descriptions (e.g., **Case 2**: specific left-right or front-back arrangements of multiple animals), the model occasionally fails to strictly adhere to rigorous geometric constraints, resulting in a semantic mismatch between the generated spatial layout and the text description.
>
> **Systematic Comparison & Robustness** To objectively evaluate robustness, we include a side-by-side comparison with current SOTA solutions in **Appendix Fig.11**. The results clearly demonstrate that:
>
> - While MOSAIC is not perfect in these extreme interaction scenarios, its performance—both in terms of subject identity preservation and spatial reconstruction—is **significantly superior** to Other methods.
>
> - Other methods typically exhibit much more severe subject confusion, significant artifacts, or complete disregard for spatial instructions in these cases.
>
> This evidence suggests that even in non-ideal scenarios, MOSAIC exhibits the strongest robustness currently available in the field.

---

> ### Author Response · Authors · 2025-11-25
> **Response to Reviewer XamC - Q2**
>
> **Response to Q2**
>
> We thank the reviewer for the inquiry regarding the fairness of the evaluation settings and the details of data distribution. To ensure impartial and unbiased results, we maintain strict control over the sample size(512 $\times$ 512), category distribution, and prompt diversity of our benchmarks. Detailed statistics are illustrated in **Appendix Fig.15**:
>
> 1. **Detailed Distribution of DreamBench (Fig.15 Left)** As a classic benchmark for personalization, DreamBench is structured to maintain a perfect **50/50 balance** between single and multi-subject scenarios in our setting:
>
>     - **Single-Subject (50%)**: Covers **Objects (35%)** and **Animals (15%)**, ensuring coverage of basic object and biological features.
>
>     - **Multi-Subject (50%)**: Evenly distributes interaction scenarios across **Animal+Animal**, **Animal+Object**, and **Object+Object** (approx. **16.7% each**).
>
>
> 2. **Detailed Distribution of XVerseBench (Fig.15 Right)** XVerseBench focuses on evaluating complex multi-subject interaction capabilities. It comprises a highly balanced test set of **300 cases**:
>
>     - **Subject Count Distribution**: covers a difficulty gradient consisting of 90 Single-Subject, **120 Dual-Subject**, and **90 Triple-Subject** samples.
>
>     - **Category Combination Diversity**: The test set covers three core categories: Human, Animal, and Object. Crucially, it explores complex permutations. For instance, in triple-subject scenarios, it include 9 distinct combination patterns (e.g., "Human+Human+Object", "Animal+Object+Object"), with 10 examples for each pattern. This prevents the evaluation from being skewed toward any specific combination (e.g., only faces).
>
> **Addressing Data Bias**: In summary, the evaluation setup achieves high balance across two key dimensions: **"Subject Count (from single to triple)"** and "**Semantic Category (Human/Object/Animal and their combinations)."** This structured design effectively eliminates data bias associated with any single domain, ensuring that MOSAIC is compared against other baselines in a fair, comprehensive, and challenging environment.

---

> ### Author Response · Authors · 2025-11-25
> **Response to Reviewer XamC - Q3**
>
> **Response to Q3**:
>
> We thank the reviewer for this question. **Yes**, we affirm that during the construction of SemAlign-MS, we prioritize not only scale but also the **balance of category distribution** and the **diversity of scene descriptions** as core construction principles.
>
> 1. **Balanced Category Distribution** As illustrated in the statistics of **Appendix Fig.12,** our dataset is not skewed toward any specific domain (e.g., exclusively faces or objects). Instead, it extensively and evenly covers three core categories: **Humans, Animals, and Generic Objects**. This cross-category balance ensures that the model learns universal semantic correspondences rather than overfitting to specific category features.
>
> 2. **Diversity of Interaction Scenes** The dataset includes a substantial amount of **"Subject-Subject"** interaction data (e.g., interactions involving **Human-Human**, **Human-Object**, **Human-Animal**, **"Animal-Object"** and so on). By incorporating these samples rich in interaction logic, we ensure diversity across the scene dimension, thereby supporting the model's ability to decouple and generate subjects within complex contexts.
>
> **Conclusion**: In summary, SemAlign-MS maintains excellent balance in both category composition and scene semantics, exhibiting **no significant distribution bias**.

---

> ### Author Response · Authors · 2025-11-25
> **Response to Reviewer XamC - Q4**
>
> **Response to Q4**:
>
> We appreciate the reviewer's suggestion for visualization. We respectfully wish to clarify a minor point regarding the mechanism: **MDL ($L_{MD}$)** and **SCAL ($L_{SCA}$)** function exclusively as **objective functions** (losses) during the training phase to guide the model in learning semantic alignment and subject decoupling. **During the inference phase, these losses are not computed**; the model simply uses the learned weights. Therefore, we cannot directly visualize the step-by-step changes of these losses during inference.
>
> **However**, to intuitively illustrate how these modules influence the generation outcome, we follow your suggestion and visualize the **Step-by-step Denoising Evolution** of models trained under three different settings. As shown in **Appendix Fig.7**:
> - Left Column: Baseline (w/o $L_{SCA}$ & $L_{MD}$).
> - Middle Column: w/ $L_{SCA}$ only (w/o $L_{MD}$).
> - Right Column: MOSAIC (Full Method).
>
> **Visual Analysis**:
> 1. **Baseline (Left)**: During the denoising process, while the model generates a "dog" and a "car," the generated car loses specific identity details present in the reference image (particularly the front design of the car). This indicates that without SCAL, the model lacks fine-grained identity preservation capabilities.
>
> 2. **w/ $L_{SCA}$ (Middle)**: With the addition of $L_{SCA}$, the identity features of the car are successfully recovered. **However, a critical observation** is that the generated dog exhibits **an unnatural yellowish tint**. This occurs because, without the decoupling constraint of $L_{MD}$, the yellow color feature from the toy car's steering wheel in the reference image erroneously "**entangles**" with the dog (Feature Entanglement/Color Bleeding).
>
> 3. **MOSAIC (Right)**: Upon introducing $L_{MD}$, this feature confusion is effectively suppressed. The generated dog restores its correct black-and-white coloration, while the yellow color is correctly confined to the toy car's steering wheel. This intuitively proves that $L_{MD}$ successfully **spatially decouples** the attributes of different subjects, preventing mutual contamination of color and texture.
>
> **Conclusion**:Through this comparative visualization, we can intuitively understand the synergy: **SCAL is responsible for "pulling" identity features (Alignment), while MDL is responsible for "pushing" away interfering features (Decoupling)**. Together, they achieve high-quality, confusion-free multi-subject generation.

---

> ### Author Response · Authors · 2025-11-25
> **Response to Reviewer XamC - Q5**
>
> **Response to Q5**:
>
> We thank the reviewer for the inquiry regarding the rationale behind the LoRA rank setting. To determine the optimal rank, we conducted a detailed ablation study to evaluate the impact of different rank sizes (32, 64, 128, 256) on the model's final performance.
>
> 1. **Ablation Results on DreamBench (Multi-Reference)** The specific experimental data is presented in the table below:
>
>     | lora rank | CLIP-I | CLIP-T | DINO |
>     | ---------- |----------|----------|----------|
>     | 32 | 74.30 | 31.55 | 54.02 |
>     | 64 | 75.71 | 32.10 | 55.90 |
>     | 128| 76.30 | 32.40 | 56.83 |
>     | 256 | 76.33 | 32.45 | 56.85 |
>
> Based on these results, we observe two distinct phases:
> - **Performance Growth Phase (Rank $32 \to 128$)**: As the rank increases, the learnable parameter capacity expands, leading to a significant upward trend across all metrics (CLIP-I, CLIP-T, DINO).
> - **Performance Saturation Phase (Rank $128 \to 256$)**: When the rank is further increased to 256, the performance gains become negligible (e.g., CLIP-I improves by only 0.03, and DINO by 0.02), showing clear **diminishing returns**.
>
> **Conclusion**:Considering the trade-off between model performance and **training costs (VRAM usage and computational load)**, we identified Rank = 128 as the optimal balance point. It ensures the model has sufficient capacity for high-quality alignment and disentanglement while avoiding unnecessary resource overhead. Therefore, we adopt 128 as the default setting for our final experiments.

---

> ### Author Response · Authors · 2025-11-25
> **Response to Reviewer XamC - Q6**
>
> **Response to Q6**:
>
> We thank the reviewer for raising this highly practical question. In real-world applications, varying quality among reference images (e.g., occlusions or resolution discrepancies) is indeed very common. Evaluating robustness in these challenging scenarios is crucial.
>
> To validate MOSAIC's performance, we design specific visualization experiments. As shown in **Appendix Fig.9**, top row demonstrate cases with **Partial Occlusion**, while the bottom row display cases with **Mixed Resolution**.
>
> 1. **Capability to Handle Partial Occlusion (Appendix Fig.9 Top Row)** In occlusion scenarios, such as a partially masked clock (left) or a woman covering her eyes with her hands (right), the reference images lose significant visual information.
>
>     - **Observation**: However, MOSAIC successfully generates complete, clear, and identity-consistent subjects.
>
>     - **Mechanism**: This indicates that the model does not merely mechanically copy reference pixels. Instead, it successfully **anchors the subject's identity using key semantic points from the unoccluded regions**, effectively completing the missing semantic information.
>
> 2. **Capability to Handle Mixed Resolution (Appendix Fig.9 Bottom Row)** In mixed resolution scenarios, such as "a high-res TV + a low-res/blurred teddy bear" (left) or "a clear face + a blurred T-shirt" (right), the input information for some subjects is highly limited.
>
>     - **Observation**: Despite this, all subjects in the generated results exhibit uniform high-quality details.
>
>     - **Mechanism**: This demonstrates that our alignment mechanism can effectively extract core semantic features even from low-quality inputs and align them within a unified feature space, ultimately producing high-quality generations.
>
> **Conclusion**: The visualization results provide strong evidence that MOSAIC maintains exceptional generation quality even when facing compromised or incomplete reference images. This is primarily attributed to the model's **robust fine-grained semantic feature alignment**. Even with imperfect inputs, our $L_{SCA}$ loss effectively guides the model to capture critical semantic correspondences to anchor identity, thereby sustaining high consistency in complex multi-subject scenes.

---

> ### Author Response · Authors · 2025-11-26
> **Kindly reminder**
>
> Dear Reviewer XamC,
>
> We hope this message finds you well. We sincerely thank you for your review of our work.
>
> We are writing to gently follow up on the rebuttal we submitted recently. We would be very grateful if you could let us know whether our response successfully addresses your concerns.
>
> We are committed to improving our work based on your guidance and stand ready to provide further clarification if required.
>
> Thank you for your time and consideration.
>
> Best regards,
>
> MOSAIC Authors

---

### Official Review · Reviewer_pkpp · 2025-11-01

**Soundness:** 3
**Presentation:** 3
**Contribution:** 2
**Rating:** 4
**Confidence:** 4

**Summary:**

The paper introduces MOSAIC, a multi-subject personalization framework that supervises reference to target attention with SCAL and reduces cross-subject leakage by MDL. It also releases SemAlign-MS, a dataset of dense point correspondences enabling effective attention-level supervision. On DreamBench and XVerseBench, MOSAIC achieves SOTA gains, for scenes with 4+ subjects.

**Strengths:**

1. The paper represent the multi-object personalization as representation optimization problem. Then introduce two different losses for the combined target. The storytelling is clear and straightforward.

2. The paper prove the 4+ object personalization tasks with less degradation. May apply for complex personalization cases.

3. The two losses introduced is well-defined and easy to follow.

**Weaknesses:**

1. The comparisons between the FLUX-based model to other baseline finetuned from SDXL may not be fair. While the increasing on performance is rather marginal. The method may works for other backbones as well since the main contribution is two of losses. It will be better to see the improvement across conventional backbones.

2. The quality of SemAlign-MS relies on the introduced construction models. It is not clearly how accurate the quality of the dataset itself.

3. Cost of computation complexity has not been clearly discussed.

**Questions:**

1. Can you report results on a second backbone (e.g., SDXL) to demonstrate novelty of SCAL/MDL?

2. How do performance and training cost scale with the number/quality of correspondence points?

3. How is quality check of the SemAlign-MS dataset?

4. Any failure cases for MOSAIC still struggles?

---

> ### Author Response · Authors · 2025-11-25
> **Response to Reviewer pkpp - W1&Q1**
>
> **Response to W1 & Q1**:
>
> We thank the reviewer for the valuable suggestions regarding the fairness of base model comparisons and the generalizability of our method.
>
> 1. **Clarification on Fairness of Existing Comparisons** First, we wish to clarify a potential misunderstanding regarding the baselines used in our main paper. The state-of-the-art (SOTA) methods compared in Table 1 and Table 2 (specifically UNO, DreamO, and XVerse) utilize the FLUX backbone, identical to our setting, rather than SDXL.Therefore, the performance advantages of MOSAIC across CLIP-I, CLIP-T, and DINO metrics directly reflect the improvements brought by our proposed $L_{SCA}$ and $L_{MD}$, rather than performance gaps stemming from different base models.
>
> 2. **Generalization to SDXL (Validation on a Second Backbone)** To fully address your concern and demonstrate that our method generalizes across different architectures (e.g., generalizing from DiT to UNet), we followe your suggestion and apply the SCAL and MDL to SDXL. We compare this "MOSAIC-SDXL" against mainstream SDXL-based methods. The results are presented below:
>
>     | Reference | Method | CLIP-I ↑ | CLIP-T ↑ | DINO ↑ |
>     |-----------|----------------|----------|----------|--------|
>     | Single-Subject | DreamBooth | 80.30 | 30.52 | 66.81 |
>     | | BLIP-Diffusion | 80.47 | 30.24 | 69.82 |
>     | | SSR-Encoder | 82.10 | 30.79 | 61.22 |
>     | | MS-Diffusion | 80.82 | 31.05 | 70.32 |
>     | | **MOSAIC-SDXL** | **82.15** | **31.20** | **72.45** |
>     | Multi-Subject | MS-Diffusion | 72.60 | 31.91 | 52.50 |
>     | | **MOSAIC-SDXL** | **73.20** | **32.11** | **53.84** |
>
>     **Analysis:**
>
>     - **Cross-Architecture Effectiveness**: When applied to the UNet-based SDXL (a non-DiT architecture), MOSAIC maintains significant effectiveness.
>
>     - **SOTA Performance**: MOSAIC-SDXL surpasses previous SDXL-based SOTA solutions (e.g., DreamBooth, MS-Diffusion) in both single-subject and multi-subject metrics.This experiment strongly validates that our contributions, $L_{SCA}$ and $L_{MD}$, are not dependent on a specific backbone but offer robust generalizability across different model architectures.

---

> ### Author Response · Authors · 2025-11-25
> **Response to Reviewer pkpp - W2&Q3**
>
> **Response to W2 & Q3**:
>
> We thank the reviewer for the important question regarding the construction quality of the SemAlign-MS dataset. To ensure high quality, we implement a strict dual-verification mechanism combining "Automated Cross-Validation" and "Human-in-the-loop Verification.", as detailed in **Appendix B.1 Stage6**
>
> 1. **Automated Cross-Validation** We do not rely on a single model. Instead, we adopt an "intersection strategy" to filter out noise. Specifically:
>
>     -  We utilize two distinct feature extraction methods: DIFT[1] and Geo-Aware[2] features.
>
>     - A sample is retained only when the corresponding points extracted by both methods are spatially highly consistent (i.e., they fall within the intersection region).
>
>     - This approach leverages the complementarity of different algorithms to automatically eliminate the vast majority of potential matching errors.
>
> 2. **Human-in-the-loop Verification** To further quantify the dataset's accuracy, we conduct a rigorous manual inspection process:
>
>     - We perform stratified sampling on 10% of the full dataset, ensuring a balanced distribution across "single/multi-reference" scenarios and various "object categories."
>
>     - Human annotators manually verified the semantic consistency of the matched points in these samples.
>
>     - The final statistics indicate that the error rate is controlled within 3%.
>
>     Through this strict automated filtering and statistical human verification, we strongly guarantee the high quality and reliability of the SemAlign-MS dataset.
>
> [1] "DIFT": Emergent Correspondence from Image Diffusion
>
> [2] "Geo-Aware": Telling Left from Right: Identifying Geometry-Aware Semantic Correspondence

---

> ### Author Response · Authors · 2025-11-25
> **Response to Reviewer pkpp - W3&Q2**
>
> **Response to W3 & Q2**:
>
> We thank the reviewer for the inquiry regarding computational complexity and scalability. We analyze this from three dimensions: performance, sampling points, and training overhead. Notably, our method introduces **no additional computational cost during inference**.
>
> 1. **Performance Scaling** As shown in **Appendix Fig.6**, we conducted a detailed ablation study on the impact of the "Sampled Number" of points on model performance:
>
>     - **Growth Phase (0-150 points)**: As the number of sampled points increases to 150, the model shows a significant upward trend in key metrics (CLIP-I, CLIP-T, DINO) on DreamBench multi-subject scenario.
>
>     - **Saturation Phase (150-200 points)**: When the points are further increased to 200, the performance gains slow down significantly (diminishing returns).
>
>     - **Decision**: Balancing performance gains with computational cost, we selecte 150 sampling points as the optimal standard.
>
>
> 2. **Training Cost Analysis** To quantify the overhead, we measure the time consumption of the SCAL and MDL under different sample numbers (see Table below). While using 150 points increases time overhead compared to 50 points, the training per-step time is only 3.68s, representing merely a **29.1% increase** in training cost compared to the baseline. However, increasing to 200 points causes a sharp spike in time cost to 4.70s, with a **64.9% increase** in training cost compared to the baseline. This confirms the rationality of setting 150 points as the default.
>
>     | Sampled Points | Training Time per Step (s) | Inference Time per Step (s)
>     |---|---|---|
>     | 0              | 2.85     | 1.53 |
>     | 50             | 3.10     | 1.53 |
>     | 100            | 3.36     | 1.53 |
>     | 150            | 3.68     | 1.53 |
>     | 200            | 4.70     | 1.53 |
>
>
> 3. **Inference Cost Analysis** We wish to emphasize that SCAL and MDL function solely as loss constraints during training and are completely removed during inference. **Therefore, MOSAIC incurs zero extra overhead at the inference stage.**
>
>     As shown in **Appendix Tab.14**, under the same Flux backbone, MOSAIC's inference speed remains consistent and outperforms both UNO and XVerse.
>
> **Conclusion**: By selecting 150 sampling points, MOSAIC achieves the best balance between training cost and performance while ensuring zero additional latency during inference.

---

> ### Author Response · Authors · 2025-11-25
> **Response to Reviewer pkpp - Q4**
>
> **Response to Q4**:
>
> We thank the reviewer for this insightful question. We firmly believe that a deep analysis of model limitations is crucial for future improvements. While MOSAIC performs remarkably well across extensive benchmarks, it still faces certain challenges when handling highly complex multi-subject interactions.
>
> As illustrated in **Appendix Fig.11**, the primary failure cases are concentrated in two types of complex scenarios:
>
> 1. **Complex Physical Containment Relationships** When the prompt describes physical containment or high overlap between objects (e.g., Case 1: "A plushie toy inside a glass jar, which is placed inside a box"), the model sometimes struggles to perfectly decouple the objects spatially. This can lead to a slight degradation in visual consistency for either the contained object (e.g., the plushie) or the container (e.g., the box).
>
> 2. **Fine-grained Spatial Relationships** In scenarios involving multiple subjects with complex relative positional descriptions (e.g., Case 2: specific left-right or front-back arrangements of multiple animals), the model occasionally fails to strictly adhere to the rigorous geometric constraints, resulting in imperfect prompt adherence.
>
> **However, it is worth emphasizing** that in **Appendix Fig.11**, we also visualize the results of current SOTA methods under the exact same prompts. The comparison clearly demonstrates that although MOSAIC is not perfect in these extreme scenarios, its performance—both in terms of subject identity preservation and spatial relationship reconstruction—is significantly superior to other baselines. Other methods often exhibit more severe subject confusion or completely ignore spatial instructions in these cases. This indicates that handling complex physical interactions remains a shared challenge in the field and serves as a key direction for our future optimization.

---

> ### Author Response · Authors · 2025-11-26
> **Kindly reminder**
>
> Dear Reviewer pkpp,
>
> We hope this message finds you well. Thank you again for your constructive feedback on our submission.
>
> We have carefully prepared our rebuttal to address the points you raised. As the discussion phase is progressing, we are eager to hear your thoughts on our response and ensure that we have met your expectations.
>
> We remain fully available to answer any further questions or provide additional details if needed. We look forward to your feedback.
>
> Sincerely,
>
> MOSAIC Authors

---

### Official Review · Reviewer_cdZj · 2025-11-01

**Soundness:** 3
**Presentation:** 3
**Contribution:** 2
**Rating:** 4
**Confidence:** 4

**Summary:**

tries to improve multi-subject personalized generation, by both aligning semantics for each subject/identity (knowing which regions in the generated image should attend to which parts of each reference), as well as disentangling each reference by pushing different subjects into orthogonal subspaces to minimize feature interference. They resolve the alignment challenge with a fine-annotated dataset and correspondence loss, and approach the later with a disentanglement loss. While both aspects are been explored and methods and datasets are proposed to resolve the challenges, MOSAIC does repurpose the similar ideas in a novel way.

**Strengths:**

1. The proposed correspondence loss for alignment is attention-based, which seems principled and interesting to me.

2. The multi-reference disentanglement loss is based on **orthogonality**, which is a fairly new perspective in multi-subject generation. This perspective is more theoretically grounded and can also inspire more theories and algorithms, the loss formulation can be considered as a novel contribution to how we enforce it in diffusion models.

3. The construction pipeline of SemAlign-MS is sound and inspires future development of new benchmark too.

**Weaknesses:**

1. SemAlign-MS lacks detailed information, e.g. I don't find number of samples, splits, distributions, etc.? To claim a dataset contribution, it is crucial to include these details, as well as examples with full annotations and comparisons with a few similar datasets.

2. Both the reference alignment and multiple-reference disentanglement have been proposed and studied. To me, the intuition and conceptual novelty is modest, e.g. [1] and its variants proposes similar alignment and [2] leverages orthogonal adaptation for multiple conditioning. I acknowledge that this work has technical novelty though.

3. Table 3: Can you add the ablation setting with MD and without SCA? This helps me compare which contributes more.

4. Can MOSAIC works for/generalizes to other datasets/tasks besides SemAlign-MS? The authors claim that MOSAIC achieves SOTA on multiple benchmarks but I don't find such experiment results. Also, I think the correspondence loss relies on fine-grained annotations, making it hard to decouple and apply to other datasets/tasks?

5. While the results look good, there lacks quantitative/qualitative evaluation of generation diversity. Because both additional losses are enhancing fidelity, it inevitably makes the training weighs identity preservation more.

[1] Xiao, Guangxuan, et al. "Fastcomposer: Tuning-free multi-subject image generation with localized attention." International Journal of Computer Vision 133.3 (2025): 1175-1194.

[2] Po, Ryan, et al. "Orthogonal adaptation for modular customization of diffusion models." Proceedings of the IEEE/CVF conference on computer vision and pattern recognition. 2024.

**Questions:**

Please see details in Weaknesses. I think this work presents new insights and shows technical novelty, but there also exists confusions and concerns regarding overclaims and experiment details.

---

> ### Author Response · Authors · 2025-11-25
> **Response to Reviewer cdZj - W1**
>
> **Response to W1**: We thank the reviewer for pointing out the need for more detailed information regarding the dataset. To substantiate our contribution claims, we add comprehensive statistics, distribution charts, annotation examples, and in-depth comparisons with other datasets in the **Appendix B.2&B.3**.
>
> 1. **Scale and Splits** As mentioned in **Line 215**(Main Paper), SemAlign-MS comprises **1.2M high-quality image pairs.**
>
>     - **Splits**: We utilize the entire 1.2M dataset for training to maximize the model's generalization capabilities. Regarding validation and testing, we strictly adhere to community standards by evaluating on independent, widely recognized benchmarks (e.g., DreamBench, XVerseBench) rather than splitting a validation set from the training distribution. This ensures a fair evaluation of the model's zero-shot generalization ability.
>
> 2. **Distributions** We add detailed statistical charts in **Appendix Fig.12** to illustrate the dataset's diversity:
>
>     - **Reference Balance**: The dataset maintains a healthy balance between single-reference and multi-reference scenarios.
>
>     - **Category Diversity**: It covers major categories such as Humans, Animals, and Objects, and includes rich cross-category combinations (e.g., Human-Object, Human-Animal interactions).
>
>     - **Prompt Complexity**: Text descriptions range from short labels to complex, long-form sentences.
>
> 3. **Full Annotation Examples In Appendix Fig.13**, we visualize specific examples of semantic point mapping. As shown, our annotations go beyond simple class labels; they provide **dense, point-wise semantic correspondences.** This density is the key enabler for achieving fine-grained alignment.
>
> 4. **Comparison with Other Datasets** In **Appendix Fig.14**, we compare SemAlign-MS with current mainstream datasets: **Subject-200K**[1] (Single-Ref) and **X2I-subject-driven**[2] (Multi-Ref):
>
>     - **More Balanced**: Compared to Subject-200K, our data is more evenly distributed across single/multi-subject scenarios.
>
>     - **Richer Combinations**: Compared to X2I-subject-driven, we include more diverse interaction combinations.
>
>     - **Crucial Novelty**: Most importantly, while previous datasets only provide image pairs or coarse masks, SemAlign-MS offers **fine-grained semantic feature point correspondences.** This is a unique property absent in prior reference datasets and serves as the foundation for our proposed Point-wise SCAL and MDL.
>
>
> [1] "OminiControl":Minimal and Universal Control for Diffusion Transformer
>
> [2] "OmniGen": Unified Image Generation

---

> ### Author Response · Authors · 2025-11-25
> **Response to Reviewer cdZj - W2**
>
> **Response to W2**:
>
> We believe there may be a misunderstanding regarding the specific scope of our contributions. While we agree that "alignment" and "disentanglement" are indeed fundamental objectives in representation learning, our core novelty lies in the granularity of our approach and the data-driven mechanism that enables it. **Crucially, we are the first to construct a comprehensive pipeline and dataset for point-wise semantic correspondences.** Different from previous works, this unique foundation allows us to achieve fine-grained feature alignment(SCAL) and spatial disentanglement(MDL) within **a unified multi-subject framework**, representing a significant leap from previous coarse-grained or optimization-based methods.
>
>
> Our method differs fundamentally from Reference [1] and [2] in three key aspects:
>
> 1. **Data-Driven Granularity: From Region to Point-level** First, our innovation is rooted in the data. Unlike previous works that rely on standard image-text pairs, we constructed the first dataset containing **Point-wise Semantic Correspondences.** This unique data foundation enables us to design a specialized **Point-wise Semantic Loss**, achieving a level of fine-grained representation control that was previously unattainable.
>
> 2. **Methodological Differences: Distinct Alignment & Disentanglement Mechanisms** We provide a detailed comparison of the implementation mechanisms:
>
> - **Comparison with FastComposer [1] (Alignment Mechanism)**:
>
>     - **[1] is Region-level**: FastComposer uses ground-truth subject masks to supervise the cross-attention maps between the prompt and latent. This achieves only a coarse, **region-aware** alignment.
>
>     - **Ours is Point-level**: Leveraging our dataset, we achieve **point-aware** alignment. By mining specific semantic matching points, we realize a much finer granularity of alignment compared to mask-based supervision.
>
> - **Comparison with Orthogonal Adaptation [2] (Disentanglement Mechanism):**
>
>     - **[2] is Parameter-level**: This method requires training **separate LoRA weights** for different subjects and enforcing orthogonality constraints between them. This increases both training complexity and parameter management overhead.
>
>     - **Ours is Attention-level**: We require only **a single global LoRA**. Our disentanglement is achieved by **spatially decoupling the attention distributions** of different subjects within the attention maps. This approach effectively prevents subject confusion while being significantly more lightweight and efficient.
>
> 3. **Unified Framework** References [1] and [2] generally focus on either alignment or disentanglement as their primary contribution. In contrast, our work leverages the semantic point information in paired data to achieve both fine-grained alignment and spatial attention disentanglement within a **single unified framework**.
>
>     **Summary of Differences:**
>     | Method | Data | Alignment | Disentanglement | Unified Framework |
>     |--------|------|-----------|-----------------|-------------------|
>     | FastComposer [1] | Standard image-text pairs | **Region-level**: Supervises cross-attention maps between subject masks and prompts | ✗ Not addressed | ✗ Alignment only |
>     | Orthogonal Adaptation [2] | Standard image-text pairs | ✗ Not addressed | **Parameter-level**: Trains separate LoRAs per subject with orthogonality constraints | ✗ Disentanglement only |
>     | **Ours** | **Point-wise semantic correspondences** (novel fine-grained dataset) | **Point-level**: Fine-grained semantic point feature matching with corresponding loss design | **Attention map-level**: Spatially decouples attention distributions of different subjects to prevent confusion | ✓ **Joint alignment and disentanglement** through semantic point matching |
>
>
> [1] "Fastcomposer: Tuning-free multi-subject image generation with localized attention."
>
> [2] "Orthogonal adaptation for modular customization of diffusion models."

---

> ### Author Response · Authors · 2025-11-25
> **Response to Reviewer cdZj - W3**
>
> **Response to W3**
>
> We thank the reviewer for this suggestion. To provide a clearer comparison of the individual contributions of the two modules, we add the ablation setting containing **only $L_{MD}$ (w/o $L_{SCA}$)** on the DreamBench multi-reference scenario. The complete ablation results are presented below:
>
> | $L_{SCA}$ | $L_{MD}$ | CLIP-I | CLIP-T | DINO |
> |-----------|----------|----------|----------|----------|
> | × | × | 73.45 | 29.90 | 52.03 |
> | √ | × | 75.89 | 31.10 | 55.99 |
> | × | √ | 75.10 | 31.70 | 55.24 |
> | √ | √ | 76.30 | 32.40 | 56.83 |
>
> Based on these results, we make the following observations:
> 1. **$L_{SCA}$ Dominates Visual Identity Preservation** Comparing the single-module results, $L_{SCA}$ yields a more significant improvement in visual metrics (**CLIP-I +2.44, DINO +3.96**) compared to $L_{MD}$ (**CLIP-I +1.65, DINO +3.21**).
>     - **Reasoning**: $L_{SCA}$ enforces strict subject-level attention alignment between the reference and generated images. This strong visual constraint directly promotes the retention of identity features.
>
> 2. **$L_{MD}$ Prioritizes Text Alignment & Disentanglement** Conversely, $L_{MD}$ contributes more significantly to **Text Alignment (CLIP-T)**, improving it by **+1.80** (compared to +1.20 from $L_{SCA}$).
>     - **Reasoning**: The core function of $L_{MD}$ is to explicitly decouple multiple subjects within the prompt (e.g., distinguishing "A golden-colored dog" from "an orange-colored cat"). Without this proper spatial decoupling, the model is prone to subject confusion or missing compositional details, which lowers prompt adherence.
>
> 3. **Synergistic Effect** When combined, the model achieves the best performance across all metrics. This confirms that "strong visual feature alignment ($L_{SCA}$)" and "spatial multi-subject decoupling ($L_{MD}$)" are complementary and essential for high-quality multi-subject generation.

---

> ### Author Response · Authors · 2025-11-25
> **Response to Reviewer cdZj - W4**
>
> **Response to W4**
>
> **W4.1: Generalization to other datasets and annotation dependency**
>
> We thank the reviewer for considering the generalizability and implementation difficulty of our method. We wish to clarify the annotation process and provide additional experiments on external datasets to alleviate your concerns.
>
> **1. Clarification on "Fine-grained Annotation" Dependency** First, we would like to address a potential misunderstanding. While MOSAIC utilizes fine-grained semantic matching points, **this does not imply a reliance on expensive or difficult-to-obtain manual annotations**.
> - **Automated Pipeline**: As described in **Appendix B.1**, we design a fully automated semantic feature point extraction pipeline. To ensure accuracy, we employed a cross-validation mechanism (combining DIFT[3] and Geo-aware[4] features) in **Appendix Line 1325**, as detailed in **Appendix B.1 Stage6**
>
> - **Reliability**: We perform human verification on a stratified sample of 10% of the data (covering single/multi-reference and various categories). The results confirms an **error rate below 3%**, demonstrating high reliability.
>
> - **Ease of Transfer**: Consequently, applying MOSAIC to other datasets (e.g., Subject200k, X2I) is straightforward. It requires only a single automated preprocessing step, eliminating any significant barrier to entry.
>
> **2. Generalization on External Datasets** To further demonstrate MOSAIC’s generalizability, we conduct supplementary experiments on two widely used public datasets: **Subject200k** (Single-Subject) and **X2I-subject-driven** (Multi-Reference).
>
> - **Performance on Subject200k (DreamBench Single-Ref)**: Since this scenario involves only single subjects, we applied our automated pipeline and enabled only $L_{SCA}$. The results show significant improvement over the baseline:
>
>     | $L_{SCA}$ | $L_{MD}$ | CLIP-I | CLIP-T | DINO |
>     |-----------|----------|----------|----------|----------|
>     | × | not-support | 81.25 | 30.05 | 74.60 |
>     | √ | not-support | **83.32** | **30.21** | **75.57** |
>
> - **Performance on X2I-subject-driven (DreamBench Multi-Ref)**: Here, we enabled both losses. The results confirm that MOSAIC effectively generalizes to external multi-subject data:
>
>     | $L_{SCA}$ | $L_{MD}$ | CLIP-I | CLIP-T | DINO |
>     |-----------|----------|----------|----------|----------|
>     | × | × | 71.30 | 29.65 | 51.73 |
>     | √ | × | 75.35 | 31.08 | 55.49 |
>     | × | √ | 74.92 | 31.60 | 55.06 |
>     | √ | √ | **75.70** | **32.15** | **56.20** |
>
> **Conclusion**: MOSAIC is not limited to the SemAlign-MS dataset. Leveraging our high-accuracy automated pipeline, the method can be easily transferred to other datasets to consistently enhance visual consistency and text control.
>
>
>
> ---
>
>
>
> **W4.2: Clarification on SOTA Experimental Results**
>
> We respectfully wish to clarify that our paper does indeed present comprehensive experimental results demonstrating that MOSAIC achieves SOTA performance across multiple benchmarks.
>
> 1. **Quantitative Comparisons**: As shown in **Table 1** (DreamBench) and **Table 2** (XVerseBench) of the main paper, we compare MOSAIC against current SOTA solutions. The data clearly indicates that MOSAIC achieves superior results on key metrics (e.g., ID-Sim, IP-Sim) in both single-reference and multi-reference scenarios.
>
> 2. **Qualitative Comparisons**: Beyond metrics, we provide extensive visual comparisons in **Fig.4** (Main Paper) and **Fig.16, Fig.17, and Fig.18** (Appendix). These visualizations intuitively demonstrate MOSAIC's significant advantage in preserving subject identity and following text instructions compared to existing methods.
>
> 3. **Reviewer Consensus**: We also note that **Reviewer XamC** reached a similar conclusion, explicitly acknowledging the SOTA performance demonstrated by our method.
>
> We sincerely invite the reviewer to revisit these tables and figures, which we believe provide strong evidence of our performance claims.
>
> [3] "DIFT": Emergent Correspondence from Image Diffusion
>
> [4] "Geo-Aware": Telling Left from Right: Identifying Geometry-Aware Semantic Correspondence

---

> ### Author Response · Authors · 2025-11-25
> **Response to Reviewer cdZj - W5**
>
> **Response to W5**:
>
> We thank the reviewer for this insightful comment. We understand the concern that the **"Fidelity vs. Diversity"** trade-off is a common challenge in personalization. To prove that our method enhances consistency without sacrificing diversity, we conducted both quantitative and qualitative assessments.
>
> 1. **Quantitative Evaluation: LPIPS Diversity Metric** We performe a rigorous "intra-class diversity" test on the DreamBench multi-reference scenario. For each test case, we fixed the prompt and generated images using **10 different random seeds**, then calculated the average **LPIPS** distance between the generated images. A higher LPIPS score indicates greater perceptual difference (better diversity).
>
>     As shown below, MOSAIC achieves the highest diversity score:
>
>     | Method | LPIPS ↑ |
>     |----------------|----------|
>     | MS-Diffusion | 45.85 |
>     | UNO | 51.20 |
>     | DreamO | 50.10 |
>     | XVerse | 48.55 |
>     | **MOSAIC** | **52.15** |
>
>     **Analysis**: MOSAIC's diversity score (52.15) is superior to existing SOTA methods. We attribute this to the nature of our **Point-wise constraint**:
>
>     - Unlike global feature alignment, $L_{SCA}$ only constrains key semantic feature points. This leaves the diffusion model with significant freedom to generate non-critical areas (backgrounds, lighting, poses).
>
>     - Simultaneously, $L_{MD}$ prevents feature entanglement between subjects, avoiding "mode collapse" where the model might otherwise revert to a confused, average representation.
>
> 2. **Qualitative Visualization** We add visualizations in **Appendix Fig.8**, showing that under fixed prompts, MOSAIC generates images with varied angles, layouts, and compositions.
>
> **Conclusion**: The results demonstrate that MOSAIC does not sacrifice diversity for fidelity; instead, it maintains competitive generation diversity while achieving SOTA-level consistency.

---

> ### Author Response · Authors · 2025-11-26
> **Kindly reminder**
>
> Dear Reviewer cdZj,
>
> We hope this message finds you well. We sincerely appreciate the time and effort you have invested in reviewing our paper.
>
> We have posted a detailed response addressing your comments and would like to kindly follow up to see if our clarifications have sufficiently resolved your concerns. We deeply value your opinion and would appreciate any feedback you might have before the discussion period ends.
>
> Please let us know if there are any remaining points you would like us to clarify.
>
> Best regards,
>
> MOSAIC Authors

---

> > ### Comment · Reviewer_cdZj · 2025-11-28
> > **Thank you for your clarification**
> >
> > Dear MOSAIC authors,
> >
> > Thank you for your detailed, dedicated response. First, I want you thank you for providing more details of the benchmark and ablation of losses, which strengthens the claimed contributions, and help me better understand the improvement from previous work. And sorry for the W4 confusion -- I overlooked end of captions for Tab 1 and 2, then it means the detailed annotation can transfer to other domains/benchmarks with automation too. For W5, I now also see why those additional losses do not make the model "overfit" to fidelity while looking like extra identify preservation. I think this is an interesting observation to extend though, and I suggest the authors briefly highlight it. Since all my concerns are addressed, I will increase my rating and recommend acceptance.

---

> > > ### Comment · Reviewer_cdZj · 2025-11-28
> > >
> > > Hi, it seems that the review editing is disabled for now due to the leakage -- I will raise later when it is resumed, just note it here that I have mentioned raising my rating from 4 to 6.

---

> > > ### Author Response · Authors · 2025-11-28
> > > **Thank you for your support and valuable insights**
> > >
> > > Dear Reviewer cdZj,
> > >
> > > We sincerely thank you for your encouraging feedback and your decision to increase the rating and recommend acceptance. We are delighted to hear that our clarifications regarding the benchmarks, ablations, and the automation pipeline (W4) have successfully addressed your concerns.
> > >
> > > We particularly appreciate your insightful comment on W5. We fully agree that the observation—that point-wise constraints enhance fidelity without causing overfitting or compromising diversity—is a valuable finding. As per your suggestion, we have explicitly highlighted this insight in the revised manuscript **(Line 512 in the main paper)** to provide future readers with a deeper understanding of this mechanism.
> > >
> > > Thank you again for your time and dedication to improving our paper.
> > >
> > > Best regards,
> > >
> > > MOSAIC Authors

---

### Author Response · Authors · 2025-11-25
**Summary of Revisions**

We sincerely thank all reviewers for their thorough and constructive reviews. We are greatly encouraged by the overwhelmingly positive feedback on our work, including the recognition of MOSAIC's superior performance in multi-subject generation and the novelty of our fine-grained alignment mechanism. In the revised manuscript, we carefully address all concerns and suggestions raised by the reviewers. The key updates include:

**[Appendix A.1.10 & A.1.11] Inference Efficiency Clarification(#pkpp-W3&Q2, #XamC-W1&Q1)**: We clarify that SCAL and MDL are training-only objectives. Therefore, MOSAIC incurs **zero additional overhead** during inference. We also provide training cost analysis to demonstrate the efficiency.

**[Appendix B.2] Dataset Transparency and Quality Control (#cdZj-Q1, pkpp-W2\&Q3, XamC-W3\&Q3, Z5Z9-W4)**: We provide comprehensive statistical analyses in **Appendix Fig. 12** (including category distribution, prompt length distribution, etc.) and detail our rigorous "Dual-Verification" pipeline, which combines automated cross-validation with human-in-the-loop verification in **Appendix B.1 Stage6**. This systematic approach ensures data balance and annotation accuracy, effectively addressing concerns regarding potential data bias.


**[Appendix A.5 & A.7] Robustness and Failure Analysis(#pkpp-Q4, XamC-W4)**: We conduct a systematic analysis of failure cases(**Appendix Fig.11**) (e.g., complex containment) and robustness tests under challenging conditions (**Appendix Fig.9**) (partial occlusion, mixed resolution, high-density crowds), proving the model's stability in non-ideal scenarios.


**[Appendix A.1.4] Generalization to SDXL Backbone(#pkpp-W1)**: We extend MOSAIC to the UNet-based SDXL architecture. The results demonstrate that our method is not limited to DiT (Flux) but generalizes well to other mainstream backbones, consistently outperforming existing SDXL-based baselines.


**[Appendix A.1.7] Comparisons with Strong Baselines(#Z5Z9-W1)**: We add rigorous comparisons against strong multi-subject methods such as FastComposer and Face-Diffuser. Quantitative results confirm MOSAIC's significant advantages in both Identity Preservation (IP) and Text Consistency (PC).


**[Appendix A.1.3] Generalization to External Dataset(#cdZj-Q4)** : We validate our method on external datasets (Subject-200K and X2I-subject-driven) using our automated pipeline. The results confirm that MOSAIC effectively generalizes to other data sources without relying on manual annotations.


**[Appendix A.1.8] VLM-based Evaluation Metrics(#Z5Z9-W2)**: To better align with human preference, we introduce VLM-based metrics (UnifiedReward and HPSv3). MOSAIC achieves SOTA scores on these perceptual quality indicators, further validating its semantic fidelity.


**Moreover, we highlight several key points:**

**The mechanism-level novelty of Point-wise Alignment.** Unlike previous region-level (Mask-based) or parameter-level (Multi-LoRA-based) approaches, MOSAIC introduces the first **Point-wise Semantic Correspondence** mechanism. This granularity enables our method to achieve precise identity anchoring without the need for complex separate encoders or heavy optimization during inference.

**The foundational contribution of SemAlign-MS.** We emphasize that our contribution goes beyond the method itself; the **SemAlign-MS dataset** fills a gap in the community by providing **the first large-scale, high-quality dataset with dense semantic point annotations**. As validated by our experiments, this data foundation is robust, unbiased, and easily transferable to other tasks.

We provide detailed responses to each reviewer below and welcome any further questions or discussions.

---

### Author Response · Authors · 2025-11-30
**Summary of Rebuttal Updates and Reviewer Interactions**

Dear Area Chairs,

We sincerely appreciate the time you are dedicating to evaluating our work following the score reset. To facilitate your review, we highlight the **significant score increases** achieved prior to the reset, alongside the **consensus on strengths** and the **common concerns addressed** during the rebuttal.

**1. Key Updates: Two Reviewers Upgraded Their Scores (Prior to Reset)**

* **Reviewer Z5Z9 (Score raised 4 $\to$ 6):**
    * Comment: "The authors have provided a strong rebuttal ... I recommend accepting this work."
    * Reason: Added strong baseline comparisons, image quality evaluations, and detailed dataset statistics.
* **Reviewer cdZj (Score raised 4 $\to$ 6):**
    * Comment: "Thank you for your detailed, dedicated response ... I will increase my rating and recommend acceptance."
    * Reason: Comprehensive responses including benchmark details, loss ablations, and clarifications on generalization/diversity concerns.

**2. Consensus on Strengths**

* Novelty: The proposed point-wise semantic alignment and disentanglement mechanism is novel and effective for multi-subject generation (pkpp, cdZj, XamC).
* SOTA Performance: The method achieves state-of-the-art results on multiple benchmarks (pkpp, Z5Z9).
* Dataset Contribution: The SemAlign-MS dataset fills a critical gap in the community with its fine-grained annotations (cdZj, Z5Z9).

**3. Addressed Common Concerns**

* **Dataset Transparency & Quality** (cdZj-W1, pkpp-W2&Q3, XamC-W2&W3&Q3, Z5Z9-W4):
    * We added comprehensive statistics (category distribution, prompt complexity) and a detailed bias analysis to prove the dataset's balance in *Appendix Fig.12*.
    * We clarified the "Dual-Verification" pipeline (Automated + Human-in-the-loop) and visualized full annotations to demonstrate accuracy, as detailed in *Appendix B.1 Stage6*.

* **Inference Efficiency Clarification**(pkpp-W3&Q2, XamC-W1&Q1):
    * **Zero-Overhead Inference:** We clarified that our losses are training-only, incurring **zero cost** during inference. We also provide training cost analysis to demonstrate the efficiency.

* **Generalization & Baselines** (pkpp-W1&Q1, cdZj-W4, Z5Z9-W1):
    * Architecture Generalization: We extended MOSAIC to SDXL (UNet), proving our method is not limited to Flux (DiT). (Response to pkpp)
    * Dataset Generalization: We validated our automated pipeline on external datasets (Subject-200K[1], X2I-subject-driven[2]). (Response to cdZj)
    * Strong Baselines: We added comparisons against strong identity-preserving methods (e.g., FastComposer[3], Face-Diffuser[4]), where MOSAIC showed significant advantages in Identity Preservation metrics. (Response to Z5Z9)

* **Robustness and Failure Analysis**(pkpp-Q4, XamC-Q6&W4)
    * Robustness: We added visualizations for challenging scenarios (*Appendix Fig.9*) (occlusion, mixed resolution), demonstrating stability. (Response to XamC)
    * Failure Analysis:  We conduct a systematic analysis of failure cases in *Appendix Fig.11*. (Response to pkpp and XamC)

**Note**: To facilitate your review, we have uploaded **a revised manuscript** with **all changes highlighted in blue.**

**Conclusion**: We have strengthened the manuscript with extensive new experiments and visualizations. Given the resolution of core concerns and the resulting endorsements from reviewers **(notably, Reviewer Z5Z9  raised the score 4 $\to$ 6, and Reviewer cdZj raised the score 4 $\to$ 6)**, we respectfully invite the AC to consider these updates and the positive consensus in the final recommendation.

Best regards,

MOSAIC Authors

[1]"OminiControl:Minimal and Universal Control for Diffusion Transformer." ICCV (2025)

[2] "OmniGen: Unified Image Generation." CVPR (2025)

[3] "Fastcomposer: Tuning-free multi-subject image generation with localized attention." IJCV (2025)

[4] "High-fidelity person-centric subject-to-image synthesis." CVPR (2024)

---

### Meta-Review · Area_Chair_zQMy · 2026-01-14

**Summary:**

This paper introduces MOSAIC, a framework for multi-subject personalized image generation. It builds a dataset SemAlign-MS to provide fine-grained semantic correspondences between multiple reference subjects and target images. Reviewers' concerns primarily focus on: 1. Dataset quality and its generalization. 2. Inference efficiency. 3. The framework's generlization to other model architectures. 4. Model's performance on challenging cases.

Overall, the rebuttal addresses most of the concerns very well. Since the final score is 6,6,6,4, and the concerns from the reviewer given score 4 are also well addressed, I recommend to accept this paper.

**Reviewer Concerns:**

The major concerns regarding the dataset quality, the inference efficiency, and the failure case analysis have been addressed convincingly.

**Reviewer Scores:**

Two reviewers raised their scores from 4 to 6 after the rebuttal.

---

### Decision · Program_Chairs · 2026-01-26

Accept (Poster)